# Functional connectivity development along the sensorimotor-association axis enhances the cortical hierarchy

Audrey C. Luo [1,2,3], Valerie J. Sydnor [1,2,3], Adam Pines[1,2,3,4],
Bart Larsen [1,2,3,5,6], Aaron F. Alexander-Bloch [2,3,7], Matthew Cieslak [1,2,3],
Sydney Covitz[1,2,3], Andrew A. Chen[8], Nathalia Bianchini Esper [9], Eric Feczko [9],
Alexandre R. Franco[9,10,11], Raquel E. Gur[2,3,7], Ruben C. Gur [2,3],
Audrey Houghton [5], Fengling Hu[12,13], Arielle S. Keller[1,2,3], Gregory Kiar [9],
Kahini Mehta[1,2,3], Giovanni A. Salum[9,14], Tinashe Tapera[1,2,3,15], Ting Xu [9],
Chenying Zhao [1,2,3,16], Taylor Salo[1,2,3], Damien A. Fair [5,6,17],
Russell T. Shinohara[12,13,18], Michael P. Milham [9,10] &
Theodore D. Satterthwaite [1,2,3,18] ✉

Human cortical maturation has been posited to be organized along the sensorimotor-association axis, a hierarchical axis of brain organization that spans from unimodal sensorimotor cortices to transmodal association cortices. Here, we investigate the hypothesis that the development of functional connectivity during childhood through adolescence conforms to the cortical hierarchy defined by the sensorimotor-association axis. We tested this pre-registered hypothesis in four large-scale, independent datasets (total $n = 3355$; ages 5–23 years): the Philadelphia Neurodevelopmental Cohort ($n = 1207$), Nathan Kline Institute-Rockland Sample ($n = 397$), Human Connectome Project: Development ($n = 625$), and Healthy Brain Network ($n = 1126$). Across datasets, the development of functional connectivity systematically varied along the sensorimotor-association axis. Connectivity in sensorimotor regions increased, whereas connectivity in association cortices declined, refining and reinforcing the cortical hierarchy. These consistent and generalizable results establish that the sensorimotor-association axis of cortical organization encodes the dominant pattern of functional connectivity development.

The mature human brain is endowed with extensive functional diversity[1–4], which gives rise to the expansive behavioral and cognitive repertoire uniquely found in humans. Such functional diversity expands during neurodevelopment. A substantial degree of this diversity is understood to be due to variation among the capabilities of sensorimotor and association cortices[5]. Sensory and motor cortices are functionally specific and support externally oriented processes, such as perception and movement. In contrast, association cortices are functionally flexible, integrative, and are recruited for both abstract cognition and internally-directed mentation[6,7]. Differences in functional capacities between sensorimotor and association cortices are thought to stem in part from regional differences in functional connectivity profiles[5], which may in turn arise from differential refinement of connectivity during brain development[6]. Mapping spatial variation in the development of functional connectivity can ultimately provide insight into how the brain's developmental program gives rise to diverse functional capacities. Here, we sought to test the

hypothesis that the development of cortico-cortical functional connectivity varies across the cortical hierarchy.

In contrast to the maturation of cortical morphology[8–10] and white matter[11,12], there is not a widely agreed-upon pattern of functional connectivity development. While there is clear variation in functional connectivity profiles across the cortex[5,13], studies characterizing how such variation arises in development have yielded inconclusive results. For instance, one study described the dorsal attention network as globally decreasing in integration[14] whereas another reported strongly increasing connectivity with sensorimotor networks[15] during development. Furthermore, different networks have been reported to have directionally opposite overall changes in functional connectivity during development, for instance default mode network (DMN) segregation[16,17] and somatomotor integration[15,16,18], without an interpretative framework that could explain such findings. Given that brain development in youth shapes brain organization in adults, one promising approach is to use known properties of hierarchical cortical organization[19,20] to contextualize developmental changes.

A major axis of hierarchical cortical organization is the sensorimotor-association (S-A) axis, which spans from primary visual and somatomotor cortices to transmodal association cortices[19]. The S-A axis describes a pattern of cortical organization that aligns with hierarchies of cortical anatomy (regional variation in feed-forward versus feed-back projections)[21,22], cortical function (from sensation to introspection)[7], and cortical evolution (degree of cortical expansion in humans)[23]. Furthermore, while the S-A axis aligns with the principal gradient of functional connectivity in the adult human brain[5], it remains unknown as to whether development of functional connectivity varies systematically along this axis. Previous studies have shown that hierarchical organization emerges gradually in development[24] and that cortical maturation may be organized along this axis[19,25]. These studies suggest that spatial variability in the development of functional connectivity may also be parsimoniously captured by the S-A axis.

Consistent with this idea, we recently reported that age-related changes in between-network coupling varied according to a network's position on the cortical hierarchy[18]. Specifically, sensorimotor networks integrated (i.e., increased connectivity) with other networks, whereas association networks segregated (i.e., decreased in connectivity) from other networks through childhood and adolescence. These findings suggest that divergent development of functional integration and segregation may occur along the S-A axis and contribute to inter-regional diversity in function.

While this earlier study included a large sample, generalizability was not evaluated as it only included data from a single site. As such, our earlier results fit within an existing literature marked by heterogeneity. This heterogeneity is a symptom of the broader reproducibility crisis in translational and developmental neuroimaging. Small samples of high-dimensional functional MRI (fMRI) data have been repeatedly shown to lead to spurious, inconsistent, or exaggerated findings[26,27]. As a result, there is an urgent need for investigators to adopt practices that have been shown to yield replicable and generalizable findings: study preregistration, independent replication samples, standardized processing pipelines, and adequately powered sample sizes[26–29]. Recent work using structural MRI has endeavored to map the trajectories of brain growth and morphometry throughout the lifespan using multiple large-scale datasets, taking an important step toward replicable human neuroscience that is generalizable to the population[29].

Here, we sought to define highly generalizable and replicable spatiotemporal patterns of functional connectivity brain development in youth. In this pre-registered study[30], we analyzed four independent, large-scale datasets (total $n = 3355$) of youth ages 5–23 years old using standardized processing pipelines[31–33]. To maximize generalizability, we sought to include data from diverse populations and different sites that were collected using disparate acquisition parameters. We hypothesized that the development of functional connectivity would diverge across the S-A axis, linking developmental variability of functional connectivity in youth to hierarchical feature variability in adulthood. Specifically, we predicted that unimodal sensorimotor cortices would exhibit increasing connectivity with age while transmodal association cortices would tend to show weakening connectivity (replicating and extending ref. 18). As described below, results define fundamental patterns of functional connectivity development across the cortical hierarchy and provide consistent evidence for an S-A axis of human cortico-cortical functional connectivity maturation.

## Results

To characterize the developmental refinement of cortico-cortical functional connectivity, we used resting-state and task-functional MRI from four independent datasets (total $n = 3355$). The Philadelphia Neurodevelopmental Cohort (PNC; $n = 1207$; ages 8–23) was used as the discovery cohort[34]. Findings were replicated in three other large-scale developmental datasets: Nathan Kline Institute-Rockland Sample (NKI; $n = 397$; ages 6–22), Human Connectome Project: Development (HCP-D; $n = 625$; ages 5–22), and Healthy Brain Network (HBN; $n = 1126$; ages 5–22)[35–37]. We examined functional connectivity (FC) strength as our primary functional connectivity metric. FC strength was quantified as the mean edge strength between a given region and all other cortical regions. Individual edge strength was characterized by the Pearson correlation between timeseries for each pair of regions. All measures were harmonized across sites in multi-site datasets[38,39].

First, we sought to establish whether the spatial distribution of developmental changes in FC strength replicated across the four datasets. Second, we investigated whether developmental changes in FC strength varied along the S-A axis. We also examined whether spatial variation in FC strength increasingly resembled the S-A axis with age. Finally, we aimed to delineate hierarchical patterns of functional segregation and integration using average between- and within-network connectivity as well as edge-level connectivity. To model linear and non-linear developmental changes in each functional connectivity metric, we fit generalized additive models (GAMs) for each brain region, with age as a smooth term and sex and in-scanner head motion as linear covariates. Our primary cortical parcellation was the Schaefer 200 atlas with a 7-network partition based on the 7 Yeo network solution[13,40]. The schematic shown in Supplementary Fig. 1 summarizes the parameter space for each dataset. All results were evaluated across multiple cortical parcellations and network partitions as sensitivity analyses (Supplementary Fig. 2).

### Replicable spatial patterns of functional connectivity development

We first sought to characterize the magnitude and direction of developmental changes in FC strength across the cortex and evaluate the extent to which these changes were similar across datasets. To quantify the overall magnitude of regional age effects, the effect size of each age spline was computed as the change in adjusted $R^2$ ($\Delta R^2_{adj}$) between a full model and reduced model with no age term. A quantitative analysis confirmed remarkably high consistency in age effects across the four independent datasets. Specifically, spatial Pearson's correlations between FC strength age effects for each pair of datasets range from 0.49 to 0.88 (mean correlation = 0.71, $p_{spin} = 0.00175$ for NKI-HBN; $p_{spin} = 0.0001$ for all other pairwise correlations; Fig. 1a). We note that correlations for NKI may potentially be lower than for other datasets due to a smaller sample size and lack of susceptibility distortion correction in this dataset.

We hypothesized that these replicable age effects of FC strength would spatially align with the sensorimotor-association (S-A) axis (Fig. 1b), which spans from primary and unimodal sensorimotor cortices to heteromodal and paralimbic transmodal association cortices.

The S-A axis rank for each cortical region represents that region's relative position along a dominant feature axis that spatially corresponds with anatomical, functional, and evolutionary hierarchies of the cortex[20]. Lower-ranking regions are involved in externally oriented perception and action whereas higher-ranking regions support higher-order cognitive, social, and emotional psychological functions[5,7]. The age effect of FC strength showed a spatial pattern across the cortical surface that was similar in all four datasets and qualitatively resembled the S-A axis (Fig. 1c). Somatomotor cortices exhibited positive age effects, indicating increases in FC strength with age, whereas association cortices displayed negative age effects, indicating decreasing FC strength through development. Taken together, these results emphasize that development of functional connectivity strength is highly generalizable across datasets.

## Development of functional connectivity varies along the S-A axis

We next sought to systematically assess the extent to which the development of functional connectivity aligns with the S-A axis. To delineate developmental trajectories of FC strength in every individual brain region, we visualized the age-smooth functions produced by each regional GAM. We hypothesized that dissociable patterns of FC strength developmental trajectories would be found along the S-A axis.

We found a spectrum of developmental trajectories that varied continuously according to a region's position on the S-A axis (Fig. 2a–d). Lower ranking regions (blue; sensorimotor pole) exhibited linear increases in FC strength, middle ranking regions (yellow; middle axis) generally showed flatter trajectories, whereas highest ranking regions (red; association pole) displayed decreasing FC strength throughout development. To further illustrate this variability, we examined developmental trajectories for large-scale networks that represented each third of the S-A axis: the somatomotor network (representing the sensorimotor end of the axis), the salience/ventral

attention network (representing the middle of the axis), and the DMN (representing the association end of the axis; Fig. 2e–p). Networks were defined using the Schaefer 7-network partition based on the Yeo 7-network solution[13,40]. Developmental trajectories for visual, dorsal attention, limbic, and fronto-parietal networks are displayed in Supplementary Fig. 3. Across all datasets, brain regions in the somatomotor network, involved in motor tasks and sensation, increased in FC strength with age and did not plateau, indicating ongoing integration with other brain regions (Fig. 2e–h). Middle-axis brain regions in the salience/ventral attention network are situated between cortices that carry out externally-oriented processes (e.g., perception) and those involved in internally-oriented processes (e.g., self-referential thought)[6]. Salience/ventral attention brain regions exhibited flatter trajectories and tended to show overall increases in FC strength with age (Fig. 2i–l). The DMN, which is linked to abstract or self-referential processing, is located in transmodal association cortex. Brain regions in this network decreased in FC strength through development (Fig. 2m–p and Supplementary Fig. 4). Developmental trajectories of FC strength were strikingly similar across all four large-scale datasets.

Given the observed differences in developmental trajectories across the S-A axis, we next sought to directly quantify the degree to which developmental effects aligned with the S-A axis. We found that age-related changes in FC strength were largely explained by a brain region's position on the S-A axis, with high replicability across all datasets (Fig. 3a–d; PNC: $r = -0.71$, $p_{spin} = 0.0001$; NKI: $r = -0.56$, $p_{spin} = 0.0001$; HCP-D; $r = -0.62$, $p_{spin} = 0.0001$; HBN: $r = -0.72$, $p_{spin} = 0.0001$). FC strength of lower-order regions became more positive with age, whereas FC strength of higher-order regions became more negative. Sensitivity analyses using resting-state fMRI data alone yielded consistent results (Supplementary Fig. 5d–f; PNC: $r = -0.68$), $p_{spin} = 0.0001$; HCP-D: $r = 0.63$, $p_{spin} = 0.0001$; HBN: $r = -0.73$, $p_{spin} = 0.0001$). Results remained consistent after additional sensitivity

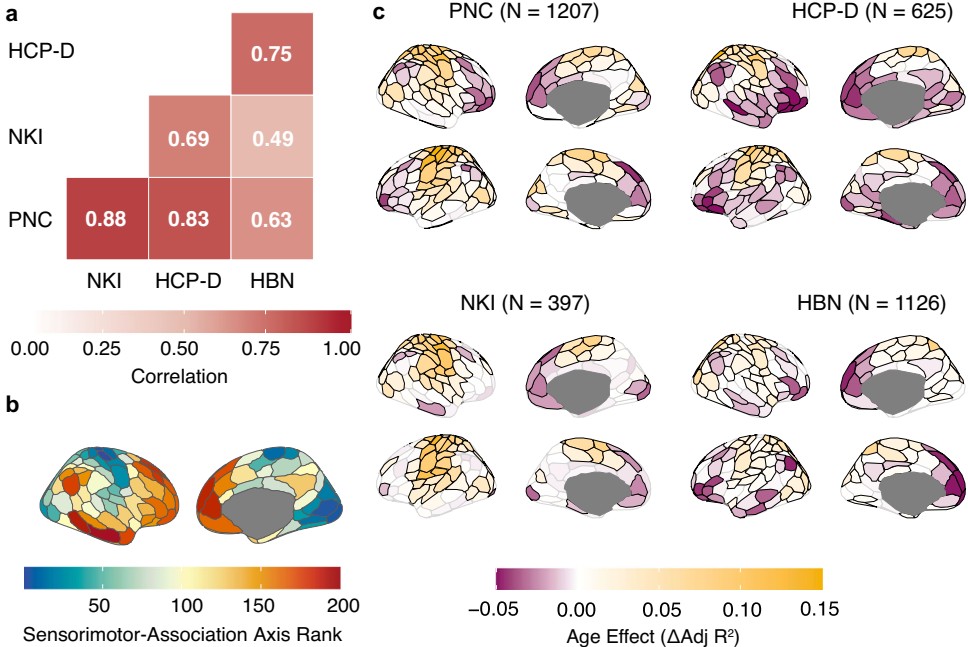

**Fig. 1 | Functional connectivity strength developmental effects replicate across four large datasets. a** The Pearson correlation plot of the age effect maps shows high spatial correlation among all four datasets ($p_{spin} = 0.00175$ for NKI-HBN; $p_{spin} = 0.0001$ for all other spatial correlations). Pearson correlation between pairs of age effect maps was used to determine spatial correlation, with statistical significance determined using spin-based spatial permutation tests. **b** The sensorimotor-association (S-A) axis is an axis of cortical organization that spans continuously from primary and unimodal sensorimotor cortices (sensorimotor

pole; dark blue; lowest ranks), to multimodal cortices (middle axis; yellow; middle ranks), and finally to transmodal association cortices (association pole; dark red; highest ranks)[19]. **c** The spatial pattern of FC strength age effects is replicable across datasets and resembles the S-A axis. Age effects are shown on the cortical surface for PNC, NKI, HCP-D, and HBN with yellow indicating increasing FC strength with age and purple indicating decreasing FC strength with age. All regions outlined in black display significant changes in FC strength ($Q_{FDR} < 0.05$). Source data are provided as a Source Data file.

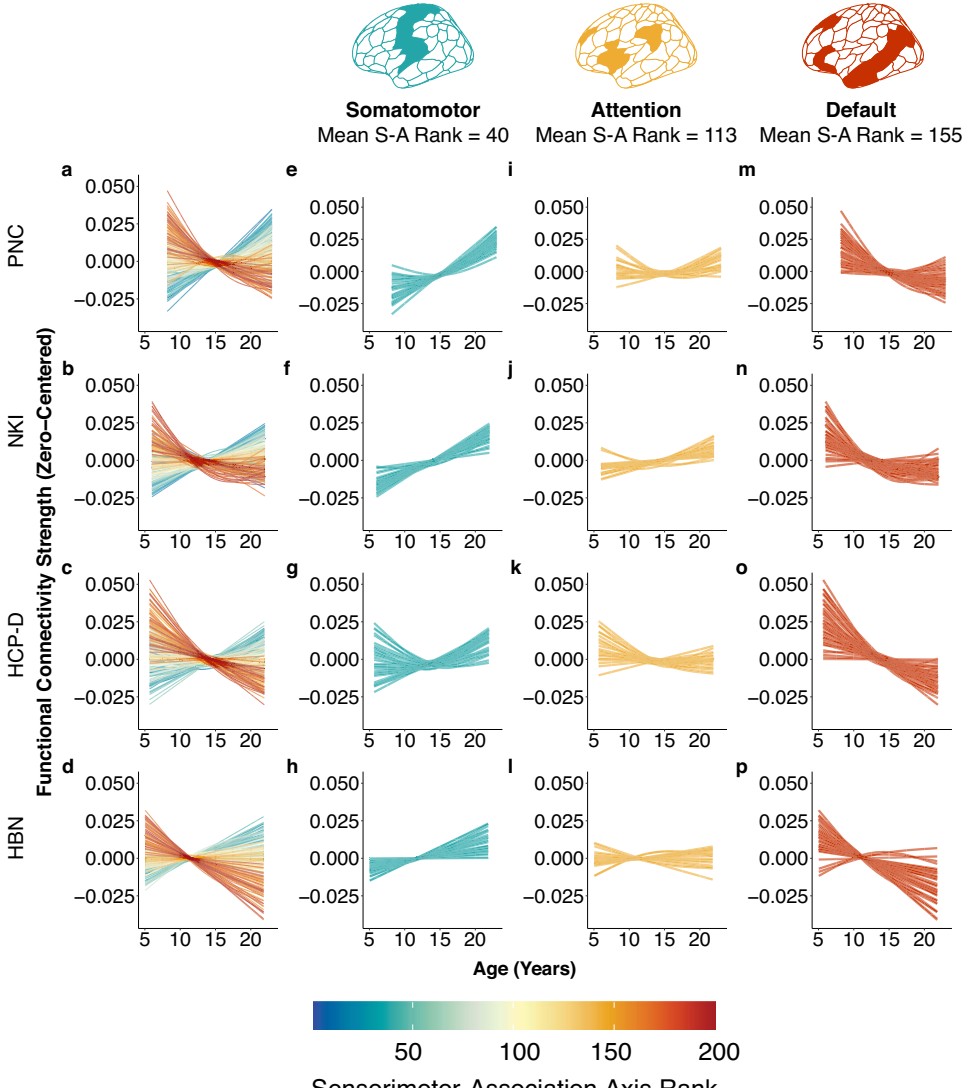

**Fig. 2 | Functional connectivity strength shows dissociable patterns of maturation across the sensorimotor-association axis. a-d** Functional connectivity strength (FC strength) developmental trajectories vary continuously along the S-A axis. Each line represents an individual region's FC strength developmental trajectory (zero-centered), modeled using generalized additive models. Colors indicate the rank of a given region along the S-A axis. **e–p** The plots display the developmental trajectories for regions from three representative functional networks: the somatomotor (**e–h**), salience/ventral attention (**i–l**), and default mode networks (**m–p**). Regions in the somatomotor network generally show increasing FC strength through development. Regions in the salience/ventral attention network exhibit both increasing and decreasing FC strength with age. Regions in the DMN predominantly show decreasing FC strength during development. Source data are provided as a Source Data file.

analyses, which computed FC strength in three additional ways: using the absolute value of the correlation coefficient as the measure of functional connectivity, thresholding connectivity matrices to only include positive correlations, and excluding global signal regression during preprocessing (Supplementary Fig. 6). These results provide consistent and generalizable evidence that FC strength development significantly and systematically varies across the S-A axis.

**Functional connectivity strength increasingly aligns with the S-A axis with age**

Our results show that FC strength age effects tend to be in opposite directions at the sensorimotor and association ends of the S-A axis, suggesting that FC strength differentiates across the axis with age. This interpretation is supported by our findings that the spatial pattern of FC strength resembled the S-A axis more at age 22 and 14 than at age 8 (Supplementary Fig. 7). As such, an outcome of this hierarchical developmental scheme may be that spatial variation in FC strength

becomes increasingly organized along the S-A axis with age. We therefore aimed to test the extent to which spatial variation in FC strength was aligned with the S-A axis throughout the course of child and adolescent development.

We performed an age-resolved analysis in which we calculated model-predicted FC strength at approximately 1-month intervals between age 5 and 22 years (as available per dataset). At each 1-month age interval, we correlated regional FC strength with S-A axis rank, producing age-specific correlation values that captured the extent to which regional differences in FC strength were accounted for by a region's location on the S-A axis. We found that the across-cortex spatial correlation between fitted FC strength and S-A axis ranks strengthened from age 5 to 22 across all datasets (Fig. 4a–d), indicating that the spatial patterning of FC strength increasingly resembled the S-A axis with age. Of note, the pattern for the NKI dataset diverges somewhat from that of other datasets, with the alignment to the S-A axis being greater in childhood. However, the pattern of FC strength

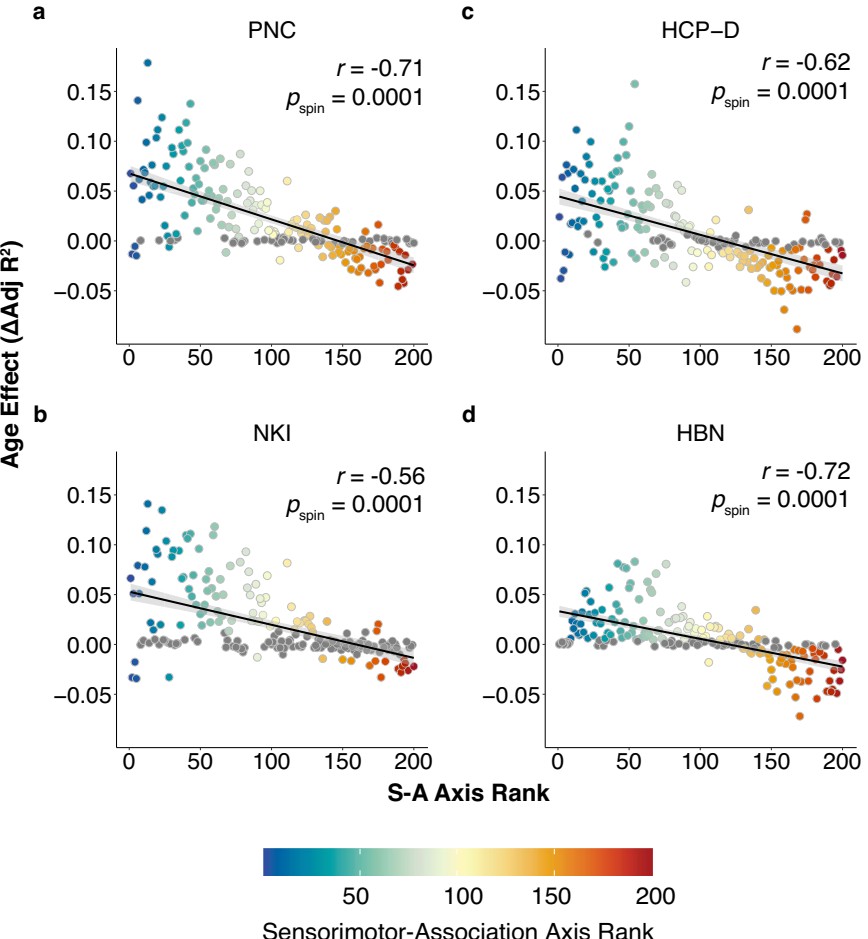

**Fig. 3 | Development of functional connectivity aligns with the sensorimotor-association axis. a** The rank of each region on the S-A axis explains the majority of variance in age effects in the Philadelphia Neurodevelopmental Cohort (PNC; $r = -0.71$, $p_{spin} = 0.0001$). These findings are replicated in additional independent datasets, including (**b**) Nathan Kline Institute-Rockland Sample (NKI; $r = -0.56$, $p_{spin} = 0.0001$), (**c**) Human Connectome Project: Development (HCP-D; $r = -0.62$, $p_{spin} = 0.0001$), and (**d**) Healthy Brain Network (HBN; $r = -0.72$, $p_{spin} = 0.0001$). The age effect of FC strength for each region (Schaefer 200) is plotted against the given region's rank on the S-A axis. Regions that do not display significant change in FC strength over development are colored in gray ($Q_{FDR} > 0.05$) and were included in the correlation. Spearman's rank correlations were used to quantify the association between S-A axis ranks and observed developmental effects, with statistical significance determined using spin-based spatial permutation tests. Source data are provided as a Source Data file.

alignment to the S-A axis converges to an absolute correlation of 0.6, consistent with other datasets. Differences may be due to NKI having a smaller sample size compared to PNC, HCP-D, and HBN and the absence of susceptibility distortion correction, leading to greater noise in the estimation of age effects in younger children. These results confirm that the S-A axis gradually captures more regional heterogeneity in cortico-cortical functional connectivity profiles throughout child and adolescent brain development.

## Hierarchical developmental changes in segregation and integration

We have found that S-A axis rank is strongly associated with how FC strength changes in development. However, while FC strength summarizes overall connectivity of a given region to the rest of the brain, it does not capture differences in connectivity based on network organization. Average between- and within-network connectivity are constituent components of FC strength and can provide insight into functional network segregation and integration, which in turn can elucidate system-specific developmental patterns of connectivity strengthening and weakening. We calculated the average between- and within-network connectivity as the mean edge strength of a given region to all other regions outside of and within that region's network,

respectively. Functional network segregation may be reflected by decreasing between-network connectivity and increasing within-network connectivity, whereas functional network integration may be represented by increases in both between-network connectivity and within-network connectivity.

First, we evaluated whether regional age effects for average between-network connectivity and within-network connectivity also varied along the cortical hierarchy. We found that both average between-network connectivity age effects and within-network connectivity age effects were associated with a region's S-A axis rank, revealing that both contributed to the developmental alignment of FC strength along the S-A axis. However, between-network connectivity developmental effects were more strongly related to the S-A axis (Fig. 5a–d; PNC: $r = -0.66$, $p_{spin} = 0.0001$; NKI: $r = -0.50$, $p_{spin} = 0.0002$; HCP-D: $r = -0.55$, $p_{spin} = 0.0001$; HBN: $r = -0.70$, $p_{spin} = 0.0001$) than within-network connectivity developmental effects (Fig. 5e–h; PNC: $r = -0.49$, $p_{spin} = 0.00085$; NKI: $r = -0.31$, $p_{spin} = 0.0335$; HCP-D: $r = -0.31$, $p_{spin} = 0.0393$; HBN: $r = -0.37$, $p_{spin} = 0.0002$). Opposite sign age effects for average between-network connectivity were seen in sensorimotor and association cortices; age effects for average within-network connectivity were largest in sensorimotor cortices. This may contribute to the

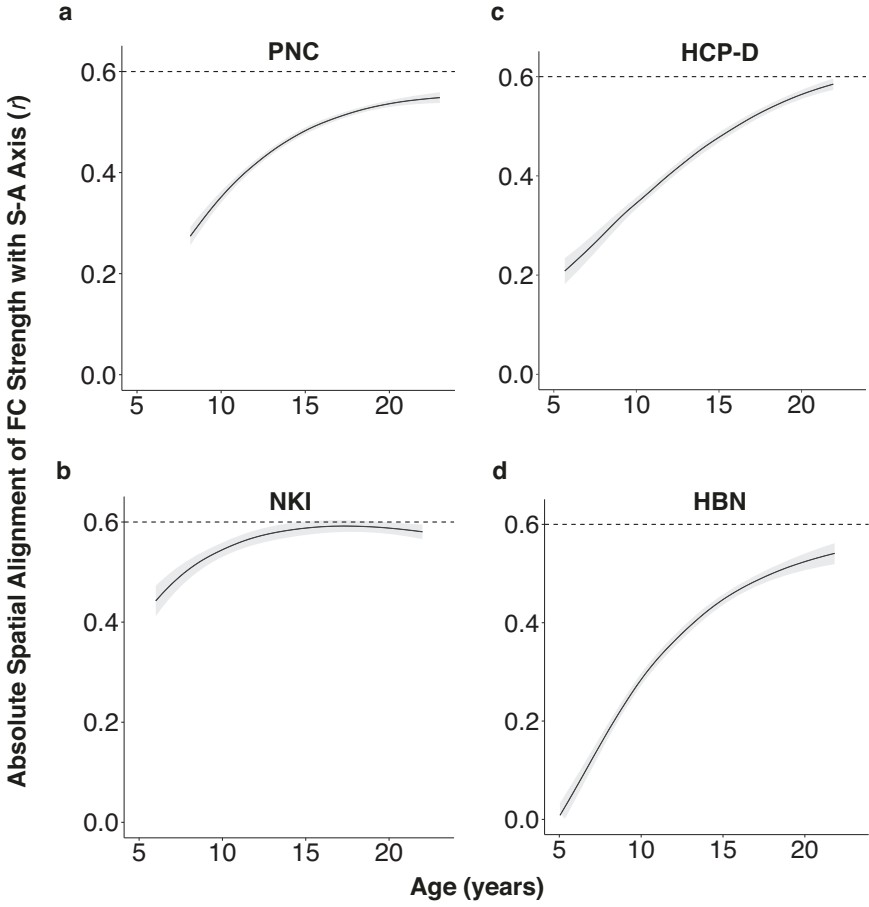

**Fig. 4 | The spatial distribution of functional connectivity strength increasingly aligns with the sensorimotor-association axis with age.** An age-resolved analysis reveals that variation in regional FC strength becomes more aligned with the S-A axis over the course of development. Plots show age-specific correlation values between FC strength and S-A axis ranks across regions, which converge to a strong absolute correlation of approximately 0.6 by early adulthood. **a** Across regions, the absolute correlation between fitted FC strength and S-A axis ranks strengthens from age 8 to 22 in the PNC. This increase in spatial alignment is replicated in (**b**) NKI (ages 6-22), (**c**) HCP-D (ages 5-22), and (**d**) HBN (ages 5–22). Model-predicted values of functional connectivity strength were generated from regional GAMs at 1-month intervals between the youngest and oldest ages in each dataset. Reliable estimates of the correlation value at each age were obtained by sampling 10,000 draws from the posterior fitted FC strength value predicted by each region's GAM smooth function. We computed age-specific correlations between fitted FC strength values across regions and S-A axis ranks for all 10,000 draws. The median correlation value (*r*) computed across all draws is represented by the black line. The 95% credible interval around the median correlation value is shown in the gray band. Source data are provided as a Source Data file.

difference in alignment of between- versus within-network effects to the S-A axis. These results suggest that low-order regions tend to integrate with other cortical regions, whereas segregation of high S-A rank regions appears to be driven by decreases in between-network connectivity rather than changes in within-network connectivity. Specifically, the lowest ranking networks, somatomotor and visual networks, tended to increase in connectivity to other networks, whereas the highest ranked network, DMN, tended to segregate from all other networks with age (Supplementary Figs. 8–10).

We next investigated the extent to which developmental changes in functional connectivity were dependent upon the identities of both brain regions forming the functional connection. To do this, we performed an edge-level analysis of functional connectivity. We modeled the effect of each combination of S-A axis rank (representing pairs of brain regions) on edge-level age effects by using a bivariate smooth interaction. Results across all datasets revealed that sensorimotor-to-sensorimotor edges tended to strengthen with age, whereas connections between sensorimotor and association regions weakened (Fig. 6a–d). Association-to-association connections also weakened with development but less prominently. Overall, these results indicate that lower S-A rank (i.e., sensorimotor) brain regions tend to integrate with other brain regions, but particularly with other low S-A rank regions. In contrast, higher S-A rank (i.e., associative) brain regions tend to segregate with age, with connections to the opposite end of the S-A axis weakening the most. Brain regions in the middle axis integrate with lower S-A rank parcels but segregate from higher S-A rank parcels.

Seed-based analyses using edge-level data examined age effects from exemplar regions from the visual, somatomotor, salience/ventral attention, fronto-parietal, and default mode networks. Visual and somatomotor regions tended to increase most in connectivity with other sensorimotor regions while decreasing in connectivity with regions in the DMN (Supplementary Figs. 11, 12). The seed from the salience/ventral attention network generally integrated with other attention network regions and with somatomotor regions but decreased in connectivity with DMN (Supplementary Fig. 13). In addition to increasing connectivity to other fronto-parietal network regions, the fronto-parietal seed tended to increase in connectivity to diverse regions across multiple networks (Supplementary Fig. 14). The DMN seed showed developmental decreases in connectivity to all other networks while increasing connectivity to other DMN regions (Supplementary Fig. 15).

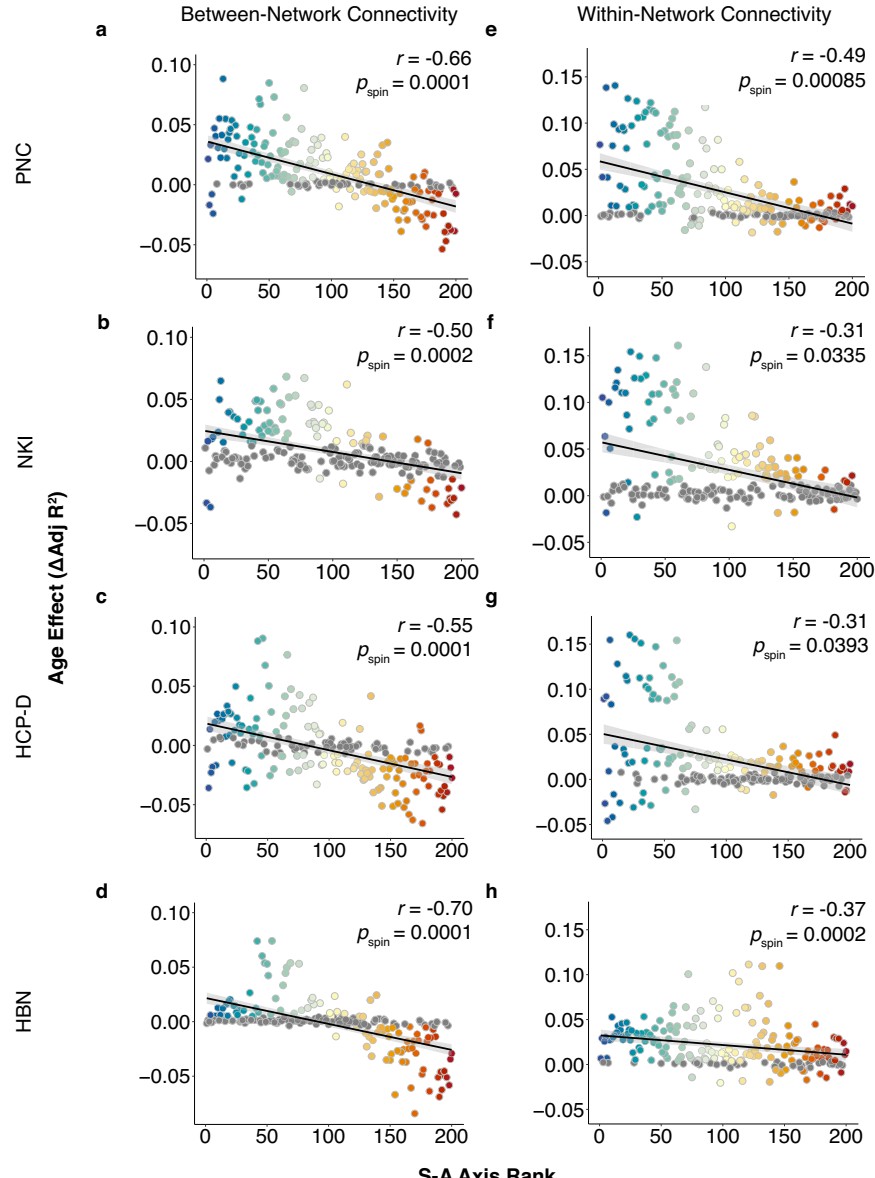

**Fig. 5 | Hierarchical developmental change in average between-network and within-network connectivity.** Regional development of average between-network connectivity and average within-network connectivity varies along the S-A axis. **a–d** The age effect of average between-network connectivity is highly aligned with the S-A axis in all four datasets (PNC: $r = -0.66$, $p_{spin} = 0.0001$; NKI: $r = -0.50$, $p_{spin} = 0.0002$; HCP-D: $r = -0.55$, $p_{spin} = 0.0001$; HBN: $r = -0.70$, $p_{spin} = 0.0001$). **e–h** The age effect of average within-network connectivity is aligned with the S-A axis in PNC ($r = -0.49$, $p_{spin} = 0.00085$) and in replication datasets (NKI: $r = -0.31$, $p_{spin} = 0.0335$; HCP-D: $r = -0.31$, $p_{spin} = 0.0393$; HBN: $r = -0.37$, $p_{spin} = 0.0002$), but

not as strongly. Results are shown in the Schaefer 200 atlas with 7-network partition based on the 7 Yeo network solution[13,40]. Colors correspond to the rank of a given region along the S-A axis. Spearman's rank correlations were used to quantify the association between S-A axis ranks and observed developmental effects, with statistical significance determined using spin-based spatial permutation tests. Note that axes were adjusted to best visualize all datasets. A total of four data points across all datasets were excluded for between-network connectivity and a total of four data points across all datasets were excluded for within-network connectivity. Source data are provided as a Source Data file.

## Discussion

We leveraged four independent, large-scale neuroimaging datasets to characterize fundamental spatial patterns of functional brain development across the cortical hierarchy. Specifically, we delineated a consistent and highly generalizable alignment between the development of functional connectivity and the S-A axis. Age-related changes in functional connectivity strength varied along this axis, with a prominent dissociation between sensorimotor and association regions at opposite poles of the axis. As a result of these hierarchically organized developmental changes, the spatial distribution of functional connectivity strength increasingly aligns with the S-A axis with age, linking

developmental variability throughout youth to cortical organization in adulthood. Together, these results resolve heterogeneous findings in the field and provide strong evidence that the S-A axis encodes the dominant pattern by which cortico-cortical functional connectivity develops in humans. Our findings underscore the promise of capitalizing upon generalizable patterns of functional brain development in future studies of human brain maturation in health and disease.

Due to inconsistencies in prior findings[41–45] and the lack of an interpretive developmental framework, no unifying description of functional connectivity maturational patterns has previously been agreed upon. For instance, both increasing[42,43,46] and decreasing[15,16]

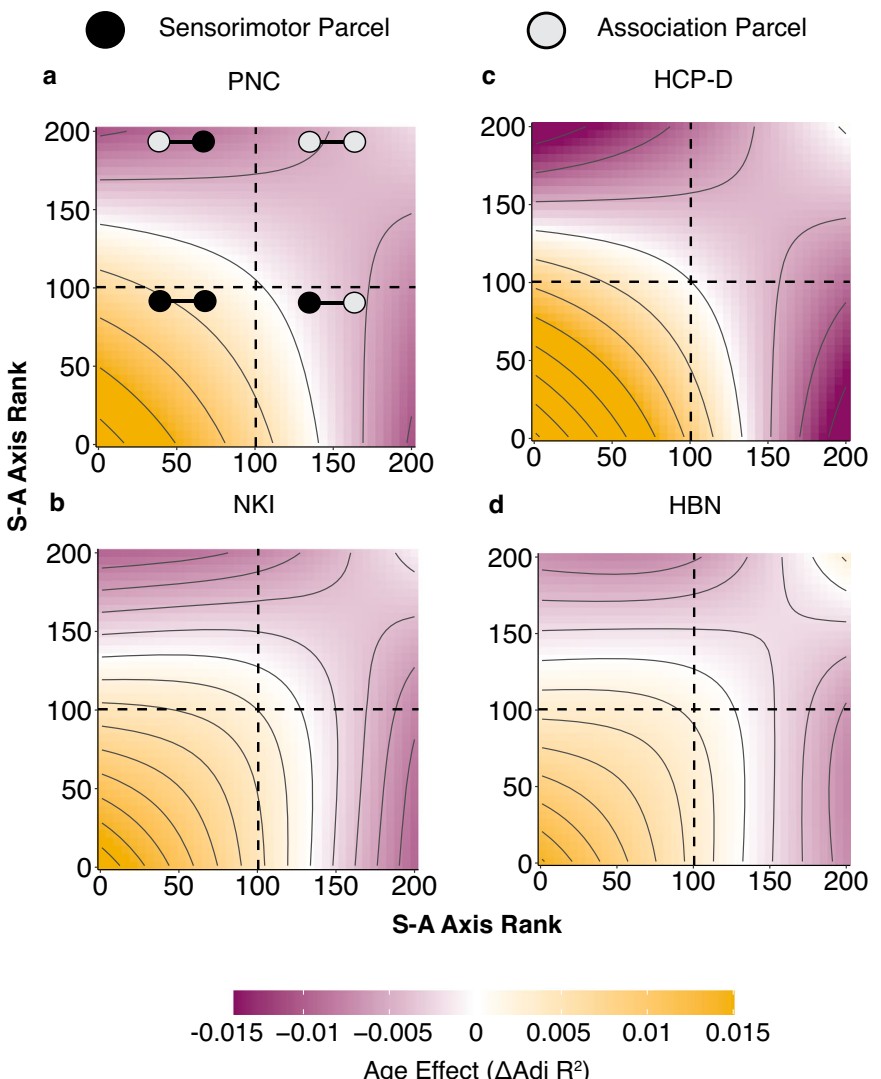

**Fig. 6 | Edge-level age effects confirm divergent connectivity refinement along the sensorimotor-association axis. a**–**d** Topographical plots display edge-level connectivity age effects as a function of S-A axis rank. Colors indicate the magnitude and direction of the age effect at each region-to-region connection. Contour lines indicate a constant age effect. Sensorimotor-to-sensorimotor connections (bottom left of each plot; yellow) display strong increases in connectivity. Connections between middle-axis regions (middle of the figure; white or light purple) demonstrate lack of change or modest decreases in connectivity. Sensorimotor-to-association connections (top left and bottom right; dark purple) show strong decreases in connectivity. Association-to-association connections (top right; light purple) also moderately weaken in development. Note that plots display the age effect of edges rather than of regions. Source data are provided as a Source Data file.

between-network connectivity have been observed in higher-order brain regions such as those involved in the default and fronto-parietal networks. Even when results are not directly contradictory, the field has lacked a coherent interpretation that accounts for such heterogeneity. While prominent earlier work has suggested that long-range connectivity strengthens and short-range connectivity weakens in development[47], this was later shown to be largely driven by motion artifact[48,49]. It should be noted that such heterogeneity regarding fundamental patterns of functional brain development is relatively unique compared to widespread agreement on the direction of developmental changes in cortical thickness, which decreases in childhood and adolescence, and white matter fractional anisotropy, which increases throughout development[11,50–54]. Here, we sought to resolve heterogeneity in the field by studying functional connectivity development using an explicit, empirically-grounded interpretive framework – that brain development conforms to and shapes hierarchical cortical organization along the S-A axis – while taking many steps to promote rigor and test the generalizability of our findings.

We recently introduced a model of human cortical development which posited that the S-A axis of brain organization also represents a major spatial and temporal axis of neurodevelopment[19]. By showing that functional connectivity refinement varies continuously across the S-A axis, our findings provide strong empirical evidence for this developmental model. Our results, which were conducted on four independent, large-scale datasets, are timely in the context of recent, urgent calls for reproducible research[26,27]. A well-documented cause of the reproducibility crisis is small sample sizes for high-dimensional fMRI data that yield low statistical power[26,28]. Recent work has also shown that false positive findings may in part be due to inconsistent and custom processing and analysis pipelines[26,55,56], leading to high analytical flexibility and the potential for selective reporting[56]. Further, generalizability of neuroimaging findings has been limited due to single-site data using similar imaging parameters[56] as well as limited demographic diversity[57]. These factors may be sources of heterogeneity and discrepancies in prior work, restricting the field's understanding of functional brain development.

We addressed these challenges by following procedures to ensure rigor and evaluate the generalizability of our findings. In addition to leveraging four large-scale, independent datasets to replicate all findings, we publicly pre-registered our hypotheses, analyses, and datasets[30]. We employed standardized, containerized, and publicly available image processing pipelines to all datasets according to our preregistration to reduce analytical flexibility, limit degrees of freedom, and minimize selective reporting[55,56]. To confirm that findings were consistent across analytical choices, we conducted sensitivity analyses using four different cortical parcellations and two different community structures. Findings were also consistent across type of MRI scan (rest-only versus concatenated task-rest fMRI), preprocessing with and without global signal regression, and ways of characterizing functional connectivity strength. Lastly, our results were consistent across differences in dataset characteristics, scanning parameters, analytical approaches, and demographics, bolstering confidence in the generalizability of these results.

Across datasets, we found that cortices at the sensorimotor pole of the S-A axis exhibited marked increases in connectivity through development. These cortices became more interconnected with other sensorimotor cortices and with middle axis regions, which include attention systems. Of note, unlike the rest of the sensorimotor cortices, the primary visual cortex showed a unique decrease in functional connectivity strength during development. This finding may reflect the need for segregated visual processing in this region. Nonetheless, our findings largely suggest increased functional integration of sensorimotor regions and facilitation of cross-system coherence[6] in the age window studied. In previous work, sensorimotor cortices have been shown to undergo segregation in infancy and early childhood[58,59]; we did not observe these effects, potentially due to the age of participants in our sample. However, the literature contains disparate accounts of how sensorimotor systems develop in later development. Studies have observed both decreasing[44] and stable[60] sensorimotor connectivity. Other studies have reported increases in sensorimotor connectivity after mid-childhood[6,18], particularly to sensorimotor and attention regions[18,61], which our findings support. Such sensorimotor integration has been shown to be negatively associated with cognition[62] and has been linked to cognitive decline in aging and neurodegenerative diseases[63–65]. Thus, our results suggest a lifespan process that may begin in adolescence.

In contrast to the developmental increase in connectivity that we observed in sensorimotor regions, we found that connectivity declined with age in higher-order association regions. We observed segregation of association regions that appeared to be driven by decreasing average between-network connectivity. While both developmental integration[42,43,46] and segregation[15,16] of association cortices have been observed in previous literature, our findings generally agree with prior work reporting that association regions such as the prefrontal cortex tend to display decreases in between-network connectivity[14] and increases in within-network connectivity[16,47,60]. Segregation may support efficient information transmission, reduce cross-modal interference, facilitate specialized processing[66–68], and has been consistently shown to be positively correlated with executive function during development[14,16,18,66,69]. The maturation of association cortices observed in this age window may occur during a critical period of protracted plasticity[70].

Lastly, we found that the middle of the S-A axis exhibited an intermediate pattern of connectivity refinement between the two poles of the S-A axis. The middle axis primarily consists of cortices involved in the salience/ventral attention and dorsal attention networks[71]. Middle-axis cortices exhibited both increasing and decreasing connectivity, with greater between-network connectivity than higher-order cortices, consistent with prior literature[14]. Specifically, middle-to-sensorimotor connections strengthened whereas middle-to-association connections attenuated with development.

Strengthened connectivity between dorsal attention systems to lower-order visual cortices has been shown to be positively associated with reasoning ability[61], possibly by facilitating top-down, goal-driven attention[6]. In contrast, connectivity between DMN and attention networks has been reported to be negatively associated with age and cognition[61,72]. Of note, while we have summarized our findings focused on three major divisions of the S-A axis (sensorimotor end; middle; association end), it is important to recognize that developmental effects and connectivity changes were continuously graded along the entire axis. Taken together, our results suggest that the spectrum of developmental connectivity refinement along the S-A axis produces heterogeneous modes of inter-regional functional connectivity, which ultimately support diverse brain functions.

Our study suggests that the different developmental programs across the cortex lead to the differentiation of association from sensorimotor connectivity profiles. Our findings are consistent with recent work that showed increasing differentiation between association and sensorimotor cortex functional connectivity profiles from childhood through adolescence[24]. We observed reduced differentiation of functional connectivity across the S-A axis in HBN, which is enriched for psychopathology compared to all other datasets, suggesting that psychopathology may be linked to differences in developmental trajectories. We also found that functional connections between association and sensorimotor regions weakened the most during development. Attenuated connectivity may support the functioning of higher-order cortices, such as those in the DMN that support internal mentation, by reducing interference from sensorimotor cortices that provide extraneous input from the environment[73]. Differentiation across the S-A axis may lead to developmental strengthening of the cortical hierarchy, whereas segregation between the regions situated at the ends of the S-A axis may facilitate perceptual decoupling in higher-order cortices.

Several limitations of this study should be noted. First, we used cross-sectional neuroimaging data, precluding our ability to examine within-person development. Future work should employ longitudinal data to characterize within-person changes in functional connectivity and alignment of developmental change with S-A axis[74]. Second, BOLD signal from fMRI is sensitive to confounding factors such as head motion, which is a major challenge when studying children and adolescents[48]. We mitigated the impact of head motion by using top-performing preprocessing pipelines and by including head motion as a covariate in all analyses[75]. Third, the age window in the present study does not capture the dramatic changes in functional connectivity that occur in very early childhood, such as the segregation of sensorimotor cortices[58]. Furthermore, because we utilized standardized group average cortical parcellations and network solutions, we were not able to evaluate whether network or community sizes changed through development. Lastly, while Pearson correlation is conventionally used to define functional connectivity[76], this similarity measure is limited by factors such as nonlinear relationships and interactions between BOLD signals. Future work might also evaluate additional measures of functional connectivity, such as partial correlation and wavelet coherence[76].

In conclusion, we provide consistent evidence from four datasets that functional connectivity is refined in development along the hierarchy defined by the S-A axis. These results strongly support the hypothesis that the S-A axis is not only an axis of brain organization, but also of brain development. Our findings also suggest that functional connectivity development refines and helps to strengthen the cortical hierarchy, with implications for functional diversity throughout the human cortex. Our findings resolve prior inconsistencies in the field and provide a broadly generalizable account of human functional brain development. This work coheres with other efforts to promote reproducibility in translational neuroimaging, such as a recent paper using large-scale structural MRI datasets to develop brain growth charts with generally stable centile scores[29]. Our highly replicable and

generalizable findings support the feasibility of creating analogous functional brain growth charts[77]. Moving forward, such generalizable patterns of functional brain development may be important for understanding not just healthy brain development, but also how deviations from normative hierarchical patterns of development may confer risk for diverse psychopathology.

## Methods

### Participants
Data were drawn from four large-scale datasets: the Philadelphia Neurodevelopmental Cohort (PNC; $n = 1207$), Nathan Kline Institute-Rockland Sample (NKI; $n = 397$), Human Connectome Project: Development (HCP-D; $n = 625$), and Healthy Brain Network (HBN; $n = 1126$).

The PNC[34] is a community sample of children and adolescents from the greater Philadelphia area collected for studying brain development, which originally included $n = 1559$ participants with fMRI data. Demographic and neuroimaging data from 1207 participants ages 8–23 from the PNC were included in this study after inclusion criteria were applied (Supplementary Fig. 16). All study procedures were approved by the Institutional Review Boards of both the University of Pennsylvania and the Children's Hospital of Philadelphia.

NKI[37] is a community-ascertained lifespan sample (ages 6–85) designed to reflect U.S. demographic distributions and included $n = 1268$ participants in the original sample. Participants ages 6–22 with demographics and neuroimaging data ($n = 397$) were included in this study. The Institutional Review Board approved this project at the Nathan Kline Institute.

HCP-D[35] is a study that aims to characterize healthy brain development in children and adolescents whose sample design parallels the demographics of youth in the U.S. Participants were recruited across four sites: University of Minnesota, Harvard University, Washington University in St. Louis, and University of California-Los Angeles. The original sample included $n = 652$ participants with fMRI data. After applying inclusion criteria, we used demographic and neuroimaging data from 625 participants ages 5–22. All study procedures were approved by a central Institutional Review Board at Washington University in St. Louis.

HBN[36] aims to characterize the phenotypic heterogeneity in developmental psychopathology and consists of a community sample of children and adolescents residing in the New York City area. Families who have concerns about psychiatric symptoms in their children were encouraged to participate through a community-referred recruitment model. Participants were scanned at four sites: Staten Island Flagship Research Center, Rutgers University Brain Imaging Center, CitiGroup Cornell Brain Imaging Center, and CUNY Advanced Science Research Center. The original sample included $n = 2255$ youth with fMRI data. Demographic and neuroimaging data from 1126 participants ages 5–22 from HBN were used in this study. The study was approved by the Chesapeake Institutional Review Board.

In all studies, written informed consent was obtained for all study participants. For participants under the age of 18, written consent was provided by legal guardians and assent was obtained from participants. Demographic information for all datasets may be found in Table 1.

### Sample construction
**Age exclusion.** For each dataset, participants ages 5-23 were included in our study. In the PNC, HCP-D, and HBN, no additional participants were excluded since all participants were within the age window studied. Data from $n = 844$ individuals were excluded from NKI's original lifespan sample of $n = 1268$ due to participants being outside the desired age window.

**Medical exclusion.** Exclusion criteria included the presence of medical conditions affecting brain function (when assessed) or gross neurological abnormalities, as well as MRI scanner contraindications. In the PNC, $n = 146$ were excluded from the sample of $n = 1559$, and $n = 21$ participants were excluded from the original sample of $n = 652$ in HCP-D. Medical exclusion data was not available for NKI and HBN.

**T1 exclusion.** We excluded low-quality T1-weighted images that did not survive manual quality assurance (when possible, based on available data). For the PNC, three highly trained raters provided manual ratings of whether images were usable or not based on artifacts. Thirty-nine participants were excluded for T1 quality in the PNC. For NKI and HBN, the Swipes for Science web application[78] was used to perform visual quality control. Raters chose to pass or fail an image based on visual inspection of the general quality of the image and the blurriness between the white and gray matter boundary[79]. An additional $n = 5$ participants were excluded from NKI due to poor T1 quality. For HBN, 586 participants were excluded for T1 quality. No additional participants were excluded in HCP-D; T1 exclusion was completed by the team that collected the data.

**fMRI motion exclusion.** We excluded task and rest fMRI scans with high in-scanner head motion, as defined as mean framewise displacement ≥0.3. Participants were excluded at this stage if all fMRI scans for a given participant failed head motion exclusion. For the PNC, an additional $n = 112$ participants were excluded for high in-scanner motion. In NKI, $n = 18$ participants were excluded; in HCP-D, $n = 2$ participants were excluded; and in HBN, $n = 354$ participants were excluded for high in-scanner head motion. In PNC, HCP-D, and HBN, task and rest scans that survived head motion exclusion were concatenated to maximize scan time. NKI collected only resting-state fMRI and was not concatenated. Note that because NKI had multiple sessions of MRI scans available, we utilized scans from the session with the greatest number of scans surviving T1 and head motion exclusion for subsequent analyses.

**Scan time exclusion.** Lastly, we excluded participants with less than 7 min of concatenated resting-state and task fMRI data. In the PNC, an additional $n = 55$ participants were excluded for having a total scan time of less than 7 min. Six additional participants were excluded in NKI; $n = 4$ participants were excluded in HCP-D; and $n = 189$ participants were excluded in HBN.

Supplementary Fig. 16 summarizes sample selection and inclusion/exclusion criteria for each dataset.

### MRI data acquisition
T1-weighted structural MRI and resting-state and task fMRI data from all four datasets were used in this present study. Imaging acquisitions for PNC, NKI, HCP-D, and HBN[34–37,80] are summarized in Supplementary Tables 1–3.

After image processing (described below), denoised time-series from resting-state and task functional MRI data were concatenated for each dataset as available. This approach was informed by studies showing that functional networks are largely similar between task and rest states and that individual variability rather than task-dependent variability accounts for the majority of variation in functional connectivity[81]. Furthermore, scan length improves reliability of functional connectivity regardless of whether the data is derived from resting-state or task scans and helps better identify individual differences[82]. The range and median timeseries length and maximum number of volumes for concatenated resting-state and task fMRI scans that survived quality control are summarized in Table 2 for each dataset.

### Image processing
Preprocessing of T1-weighted images and functional MRI timeseries used fMRIPrep 20.2.3 (PNC and NKI) and 22.0.2 (HCP-D and HBN). A

**Table 1 | Demographic characteristics for each dataset**

| Dataset | N | Female (%) | Age Range (Mean, SD) | Race (self-reported) | | | | |
| --- | --- | --- | --- | --- | --- | --- | --- | --- |
| | | | | Asian | Black | Other/Mixed | White | Missing |
| PNC | 1207 | 646 (53.5%) | 8–23 (15.4 ± 3.5) | 11 (0.9%) | 513 (42.5%) | 132 (10.9%) | 551 (45.7%) | 0 (0%) |
| NKI | 397 | 186 (46.9%) | 6–22 (14.5 ± 4.4) | 34 (8.5%) | 82 (20.7%) | 11 (2.8%) | 258 (65.0%) | 12 (3.0%) |
| HCP-D | 625 | 337 (53.9%) | 5–22 (14.5 ± 4.1) | 48 (7.7%) | 69 (11%) | 97 (15.5%) | 395 (63.2%) | 16 (2.6%) |
| HBN | 1126 | 439 (40%) | 5–22 (11.6 ± 3.5) | 33 (2.9%) | 139 (12.3%) | 307 (27.3%) | 498 (44.2%) | 149 (13.2%) |

The Philadelphia Neurodevelopmental Cohort (PNC) served as the discovery dataset. Replication datasets included the Nathan-Kline Institute-Rockland Sample (NKI), the Human Connectome Project: Development (HCP-D), and the Healthy Brain Network (HBN). Demographic data on race was self-reported. The racial category "Other/Mixed" includes individuals identifying with more than one race, as well as individuals identifying as American Indian or Alaska Native, Hispanic or Latino, or Native Hawaiian or Other Pacific Islander.

newer release of fMRIPrep was used for HCP-D and HBN to allow for top up-based susceptibility distortion correction given the acquisition of reverse phase encoding directions. Following pre-processing with fMRIPrep, post-processing used XCP-D[31–33,83].

**Structural data preprocessing.** Structural images underwent correction for intensity non-uniformity with N4BiasFieldCorrection from ANTs 2.3.3[84,85], skull-stripping with a Nipype 1.6.1 implementation of ANTs brain extraction workflow, and brain tissue segmentation with fast FSL 5.0.9 (PNC and NKI) and 6.0.5.1 (HCP-D and HBN)[86]. Brain surfaces were then reconstructed using FreeSurfer 6.0.1 (PNC and NKI) and 7.2.0 (HCP-D and HBN)[87]. Volume-based spatial normalization of the T1-weighted image to two standard spaces (MNI152NLin6Asym, MNI152NLin2009cAsym) was performed through nonlinear registration with ANTs.

**Functional data preprocessing.** A skull-stripped reference BOLD volume was generated through fMRIPrep. A B0 field map was then estimated based on a phase-difference map calculated with a dual-echo GRE (gradient-recall echo) sequence (in PNC) or was estimated based on two or more echo-planar imaging (EPI) references with top up from FSL and aligned with rigid-registration to the target EPI reference run (HBN and HCP-D)[88]. The phase-difference B0 field map in PNC was converted to a displacements field map with FSL's fugue and SDCflows tools. Susceptibility distortion correction (SDC) was omitted in all NKI participants, six participants (13 scans) in HBN, and 34 participants (72 scans) in PNC as these participants did not have fieldmaps.

The BOLD reference was then co-registered with rigid transformations (six degrees of freedom) to the T1-weighted reference using bbregister in FreeSurfer. Head-motion parameters with respect to the BOLD reference were calculated before any spatiotemporal filtering using FSL's mcflirt[89]. BOLD runs were slice-time corrected using 3dTshift from AFNI 20160207[90] and resampled onto their original, native space by applying a single, composite transform to correct for head-motion and susceptibility distortions. The BOLD time-series were also resampled onto the fsaverage surface and into standard space, generating a preprocessed BOLD run in MNI152NLin6Asym space. Furthermore, to project BOLD timeseries onto the fsLR cortical surface, grayordinate files[91] containing 91k samples (32k vertices per hemisphere) were generated using the highest-resolution fsaverage as the intermediate standardized surface space.

**Functional data postprocessing.** Outputs of fMRIPrep were post-processed by XCP-D 0.0.8 (NKI), 0.3.0 (HCP-D), and 0.3.2 (PNC and HBN). XCP-D[83] is an extension of the eXtensible Connectivity Pipeline Engine (XCP)[31,32] and was developed to mitigate motion-related artifacts and noise in functional MRI data from developmental cohorts. First, outlier detection was performed. In order to identify high-motion outlier volumes, framewise displacement was calculated[92] with a head radius of 50 mm. Then, the BOLD data were despiked, mean-centered, and linearly detrended. Despiking is a temporal censoring operation

that performs similarly to scrubbing in prior benchmarking studies[75]. Thirty-six confounds were estimated based from the preprocessed timeseries in fMRIPrep: six motion parameters, mean global signal, mean white matter signal, mean CSF signal with their temporal derivatives, and the quadratic expansion of six motion parameters, tissues signals and their temporal derivatives[31,75]. The 36 nuisance regressors were regressed from the BOLD data using linear regression as implemented in Scikit-Learn 0.24.2 (NKI), Scikit-Learn 1.1.3 (HCP-D), or nilearn 0.9.2 (PNC and HBN). Processed functional timeseries were extracted from residual BOLD using Connectome Workbench[91] for the following atlases: the Schaefer 17-network 200 and 400 parcel atlas[40], the HCP-MMP atlas[93], and the Gordon atlas[94]. The Schaefer 200 atlas was used as the primary atlas and Schaefer 400, HCP-MMP, and Gordon atlases were used in sensitivity analyses. Lastly, parcellated rest and task fMRI timeseries were concatenated and the Pearson correlation between concatenated timeseries was computed for every pair of cortical regions.

## Quantification of functional connectivity metrics
To examine developmental changes in global functional connectivity profiles, we computed functional connectivity (FC) strength as our primary measure of interest. Furthermore, to gain insight into developmental changes in functional segregation and integration, we quantified average between- and within-network connectivity and edge-level connectivity as secondary measures.

To calculate the FC strength for a given cortical region, we first computed Pearson correlations of its time series with that of all other regions. We then averaged the Pearson correlations to define FC strength for that region. Hence, FC strength represents the mean edge strength of a given region with all other regions, without thresholding. Average between-network connectivity was defined as the mean edge strength (Pearson correlation) of a given region and all other regions not in that region's network. Average within-network connectivity was defined as the mean edge strength (Pearson correlation) of a given region and all other regions within that region's network. Furthermore, to examine the development of connectivity between specific pairs of functional networks, we computed the average connectivity between each pair of networks as well as within-network connectivity (derived from Yeo's 7-network solution). Of note, the average and range of S-A ranks for each network are as follows: visual (rank = 33, range = 2–109), somatomotor (rank = 40, range = 1–94), dorsal attention (rank = 84, range = 42–139), salience/ventral attention (rank = 113, range = 70–175), limbic (rank = 136, range = 108–167), fronto-parietal (rank = 144, range = 68–197), and default (rank = 155, range = 81–200).

Lastly, we examined functional connectivity at the edge level by extracting the Pearson correlation between timeseries for each pair of regions. Seed-based analyses examining age-related connectivity in exemplar regions from the visual, somatomotor, salience/ventral attention, fronto-parietal, and default mode networks were conducted using edge-level connectivity data.

**Table 2 | fMRI time-series length for each dataset**

| Dataset | Range (min) | Median timeseries length (min) | Maximum number of volumes |
|---------|-------------|-------------------------------|---------------------------|
| PNC | 8.35–33.25 | 28.25 | 665 |
| NKI | 7.75–24.10 | 24.10 | 1424 |
| HCP-D | 7.47–42.67 | 42.67 | 3200 |
| HBN | 8.33–23.33 | 18.33 | 1750 |

For each dataset, resting-state and task fMRI scans were concatenated after undergoing image processing and T1 and head motion quality control. The range and median timeseries length and maximum number of volumes for each dataset are shown here. Note that NKI only obtained resting-state fMRI and thus only rest scans were concatenated.

## Harmonization of imaging data

Harmonization of functional connectivity metrics in multi-site data (HCP-D and HBN) was performed to ensure the imaging measures were comparable across sites[95–97]. To do so, we applied an extension of Correcting Covariance Batch Effects (CovBat) where the biological covariate of age was modeled as a smooth term via a generalized additive model in both the initial mean-correction stage and the covariance-correction stage, similar to ComBat-GAM[38,39]. Harmonization was completed using the *CombatFamily* package (version 0.1.0) in R. Sex and in-scanner motion were included as covariates as well. This algorithm was chosen to remove any covariance-related batch effects that have been shown to be present in functional connectivity data while simultaneously respecting the non-linear downstream modeling approach.

## Developmental models

To model both linear and non-linear associations between functional connectivity metrics and age, GAM were fit using the *mgcv* package (version 1.8.39) in R[98–102]. GAMs were fit for each parcellated cortical region with a given functional connectivity metric (e.g., FC strength) as the dependent variable, age as a smooth term, and both sex and in-scanner motion as linear covariates:

$$Connectivity \sim s(age) + \beta_{sex} + \beta_{headmotion} \qquad (1)$$

In-scanner head motion was quantified as the mean framewise displacement averaged across all functional runs included for each subject. That is, the average mean framewise displacement of the concatenated task and rest scans surviving T1 and head motion exclusion was used as a covariate. Age was modeled using thin plate regression splines as the smooth term basis set with the maximum basis complexity ($k$) set to 3 to avoid overfitting. This basis complexity consistently resulted in the lowest model Akaike information criterion across cortical regions and datasets. The GAM smooth term for age produces a smooth function (or spline) resulting from a linear combination of weighted basis functions. This spline represents a given region's developmental trajectory for a functional connectivity metric. To examine the spatial distribution of FC strength at specific ages, we generated fitted values of FC strength from the GAM at ages 8, 14, and 22 using the "fitted_values" function in the *gratia* package (version 0.7.0).

To quantify the age effect as in prior work[18], for each brain region or edge, the effect size of age-related change was quantified by the change in adjusted $R^2$ ($\Delta R^2_{adj}$) between a full model and reduced model with no age term. The significance of the association between the functional connectivity metric and age was assessed using analysis of variance to compare the full and reduced models. To characterize the direction of the effect (increasing or decreasing functional connectivity with age), we evaluated the sign of the age coefficient from an equivalent linear model[18,103]. Multiple comparisons were controlled for with false discovery rate (FDR) correction; $Q < 0.05$. All statistical analyses were conducted in R 4.1.2.

## Correspondence of developmental effects to the S-A axis

**Alignment of functional connectivity metric age effects to the S-A axis.** To quantify the association between S-A axis ranks and observed developmental effects, we used Spearman's rank correlations. The S-A axis was derived in ref. 19 from multi-modal brain maps that exhibit stereotyped feature variability between primary sensorimotor and transmodal association cortices. As a result, the S-A axis represents the average cortical hierarchy derived from multivariate brain properties. A vertex-level S-A axis (in the fsLR surface-density 32k) was parcellated with study atlases to yield regional S-A axis ranks. The S-A axis is publicly available and was obtained from https://github.com/PennLINC/S-A_ArchetypalAxis for this study.

Specifically, ten cortical maps were used to derive the S-A axis. These ten maps included anatomical hierarchy as quantified by the T1-weighted to T2-weighted ratio[104], functional hierarchy[5], evolutionary hierarchy[105], allometric scaling[106], aerobic glycolysis[107], cerebral blood flow[108], gene expression[109], first principal component of NeuroSynth terms[110], externopyramidization[111], and cortical thickness[19]. Vertex-level representations for each of the ten maps were obtained. Vertices were rank-ordered based on feature value. Vertex rankings were then averaged across all ten maps to derive the spatial patterning of feature variability along a unidimensional S-A axis. The vertex-level S-A axis was parcellated into four cortical atlases used in this paper (Schaefer 200, Schaefer 400, HCP-MMP, and Gordon) such that each region was assigned a unique S-A rank indicating its relative position on the S-A axis.

After comparing the two parcellated cortical feature maps (i.e., S-A axis and age effect maps) using Spearman's correlation, we tested for statistical significance using spin-based spatial permutation tests using the "rotate_parcellation" algorithm in R[112]. The spin-based spatial permutation test, or "spin test," mitigates issues with distance-dependent spatial autocorrelation that is prominent in neuroimaging data. The spin test generates a null distribution by rotating spherical projections of one feature map at the cortical surface. This approach preserves the spatial covariance structure of the data[113]. Here, we generated a null distribution based on 10,000 spherical rotations. The spin tests compute a $p$ value ($p_{spin}$) by comparing the empirically observed correlation to the null.

We additionally investigated how the S-A rank of the cortical regions on each end of a functional connection is associated with developmental strengthening or weakening of the connection. To evaluate how the development of edge-level connectivity differs across the sensorimotor-association axis, we examined age-related changes in connectivity across edges by fitting a bivariate smooth interaction. The effect of S-A axis rank on edge-level age effects was modeled using a tensor product smooth to create topographical plots[18,114]:

$$\Delta R^2 adj \sim te(SA.rank_{parcel1}, SA.rank_{parcel2}) \qquad (2)$$

**Age-resolved analysis of FC strength alignment with the S-A axis.** Lastly, we performed an age-resolved analysis to examine how the spatial distribution of FC strength aligns with the S-A axis across the broad age range studied. This analysis was performed to gain insight into whether the spatial patterning of functional connectivity across the cortical mantle becomes increasingly hierarchical through development.

We first computed smooth functions from the GAM model for each region as described above. We then calculated the model-predicted FC strength at ~1-month intervals between age 5 and 22 years (as available per dataset), which corresponds to 200 unique ages. The values of FC strength across the cortex at each age was then correlated with the S-A axis rank of each brain region, quantifying the relationship between a region's FC strength and its position on the S-A axis and yielding age-specific correlations across the entire age window.

To determine a point estimate and 95% credible interval for age-specific correlation values, we used a Bayesian approach. In this approach, we first created a multivariate normal distribution based on the normal distributions of each covariate's coefficients. We then sampled from this posterior distribution 10,000 times to estimate uncertainty around the model parameters, fitted FC strength values, and ultimately the FC strength-to-SA-axis correlation value to generate credible intervals at each age. Specifically, using the posterior distribution of each region's fitted GAM, we took 10,000 draws to generate 10,000 simulated age smooth functions and fitted values of FC strength for each region. For each draw, we correlated the fitted value of FC strength with S-A axis rank at each of the 200 sampled ages. This generated a distribution of correlation values at each age, which was then used to determine the median correlation value and 95% credible interval of the correlation values for each age.

### Sensitivity analysis

To investigate whether our findings were consistent across analytic choices, such as type of MRI scan (concatenated task and rest scans versus rest only) and atlas used for cortical parcellation, sensitivity analyses were performed. First, for a sensitivity analysis using only resting-state data while excluding fMRI acquired during task conditions, PNC, HCP-D, and HBN were analyzed. Main analyses for NKI were completed with only resting-state fMRI due to the absence of task scans, and thus NKI was not included in this sensitivity analysis. We included participants with at least 6 min of resting-state fMRI. We analyzed data from 998 participants (549 females) from PNC, 611 participants (328 females) from HCP-D, and 842 participants (342 females) from HBN. The maximum scan time for resting-state scans was 11.2 min (224 volumes) for PNC, 25.5 min (1912 volumes) for HCP-D, and 10.15 min (750 volumes) for HBN.

Furthermore, analyses were evaluated using additional cortical parcellations. Our primary parcellation utilized the Schaefer 200 atlas; secondary atlases included the Schaefer 400 atlas, the Gordon atlas, and the HCP-MMP atlas[40,93,94]. For analyses of secondary outcome measures that require community structure, namely average between- and within-network connectivity, we evaluated both the Yeo 7 and 17-network partitions associated with the Schaefer atlas.

To evaluate whether the main findings with FC strength were impacted by negative correlations and global signal regression (GSR), FC strength was computed in three additional ways. First, the absolute value of the correlation coefficient was used as the measure of functional connectivity following preprocessing with GSR. Second, after processing the data using GSR, connectivity matrices were thresholded to include only positive correlations. FC strength was then computed on thresholded matrices. Thirdly, the data was processed without GSR, and signed edges were retained without thresholding. FC strength was then computed on connectivity matrices that were derived from this data.

### Reporting summary

Further information on research design is available in the Nature Portfolio Reporting Summary linked to this article.

## Data availability

This paper analyzes publicly available data from four datasets: the Philadelphia Neurodevelopmental Cohort, accessible from the Database of Genotypes and Phenotypes (phs000607.v3.p2) at https://www.ncbi.nlm.nih.gov/projects/gap/cgi-bin/study.cgi?study_id=phs000607.v3.p2; Nathan Kline Institute-Rockland Sample is available at https://openneuro.org/datasets/ds001021/versions/ 1.0.0; Human Connectome Project: Development is available for download through the NIMH Data Archive (https://nda.nih.gov/); and Healthy Brain Network is accessible through https://fcon1000.projects.nitrc.org/indi/cmihealthybrainnetwork/. Source Data[115] generated in this study have been deposited in the Zenodo database under accession https://doi.org/10.5281/zenodo.10818786.

## Code availability

Analysis code[116] is available at https://github.com/PennLINC/network_replication. A detailed description and guide to the code can be found at https://pennlinc.github.io/network_replication/.

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

## Acknowledgements

This study was supported by grants from the National Institute of Health: R01MH120482 (T.D.S. and M.P.M.), R01MH113550 (T.D.S.), R01MH112847 (R.T.S. and T.D.S.), R01EB022573 (T.D.S.), RF1MH116920 (T.D.S.), R37MH125829 (T.D.S. and D.A.F.), R01MH119219 (R.E.G. and R.C.G.), R01MH123550 (R.T.S.), R01NS060910 (R.T.S.), R01MH123563 (R.T.S.), R00MH127293 (B.L.), 5T32NS091006-08 (A.S.K.), 5T32MH019112-32 (A.S.K.), T32MH016804 (V.J.S.), F31MH123063 (A.P.), and T32GM007170 (F.H.). V.J.S. was additionally supported by a National Science Foundation Graduate Research Fellowship (DGE-1845298). Additional support was provided by NIH 1U24NS130411, NIH RF1MH121867, the AE Foundation, the Center for Artificial Intelligence and Data Science for Integrated Diagnostics at Penn, and the Penn/CHOP Lifespan Brain Institute.

## Author contributions

The study was designed by A.C.L. with V.J.S., A.P., and T.D.S. R.E.G., R.C.G. and T.D.S. provided resources and supervised collection of PNC. A.R.F., T.X., G.A.S., G.K. and M.P.M. provided resources and supervised collection of NKI and HBN. D.A.F. and E.F. provided resources for HCP-D. A.H. curated HCP-D. N.B.E., A.R.F., and T.D.S. curated and quality-checked the neuroimaging data. M.C., S.C., T.S., T.T. and K.M. processed the neuroimaging data with software tools developed by M.C., S.C., T.S., T.T. and K.M. A.C.L. implemented all statistical analyses with R code written by A.C.L., adapting several functions written by V.J.S. and B.L. A.F.A-B., R.T.S., A.A.C. and F.H. provided input on statistical approaches. V.J.S. conducted an internal code review and replication of all study findings. A.C.L. generated all figures. A.C.L. wrote the original draft and all authors (A.C.L., V.J.S., A.P., B.L., A.F.A-B., M.C., S.C., A.A.C., N.B.E., E.F., A.R.F., R.E.G., R.C.G., A.H., F.H., A.S.K., G.K., K.M., G.A.S., T.T., T.X., C.Z., D.A.F., T.S., R.T.S., M.P.M. and T.D.S.) reviewed and revised the final draft.

## Competing interests

The authors declare the following competing interest: R.T.S. reports consulting income from Octave Biosciences and has received compensation for scientific reviewing from the American Medical Association for work unrelated to the present research. All other authors declare no competing interests.

## Additional information

[1]Penn Lifespan Informatics and Neuroimaging Center (PennLINC), Perelman School of Medicine, University of Pennsylvania, Philadelphia, PA 19104, USA. [2]Department of Psychiatry, Perelman School of Medicine, University of Pennsylvania, Philadelphia, PA 19104, USA. [3]Lifespan Brain Institute, Children's Hospital of Philadelphia and Perelman School of Medicine, University of Pennsylvania, Philadelphia, PA 19104, USA. [4]Department of Psychiatry and Behavioral Sciences, Stanford University, Stanford, CA 94305, USA. [5]Masonic Institute for the Developing Brain, University of Minnesota, Minneapolis, MN 55455, USA. [6]Department of Pediatrics, University of Minnesota Medical School, Minneapolis, MN 55455, USA. [7]Department of Child and Adolescent Psychiatry and Behavioral Science, Children's Hospital of Philadelphia, Philadelphia, PA 19104, USA. [8]Division of Biostatistics and Bioinformatics, Department of Public Health Sciences, Medical University of South Carolina, Charleston, SC 29425, USA. [9]Center for the Developing Brain, Child Mind Institute, New York, NY 10022, USA. [10]Center for Biomedical Imaging and Neuromodulation, Nathan Kline Institute for Psychiatric Research, Orangeburg, NY 10962, USA. [11]Department of Psychiatry, NYU Grossman School of Medicine, New York, NY 10016, USA. [12]Penn Statistics in Imaging and Visualization Center, Perelman School of Medicine, University of Pennsylvania, Philadelphia, PA 19104, USA. [13]Department of Biostatistics, Epidemiology and Informatics, Perelman School of Medicine, University of Pennsylvania, Philadelphia, PA, USA. [14]Section on Negative Affect and Social Processes, Hospital de Clínicas de Porto Alegre, Universidade Federal do Rio Grande do Sul, Porto Alegre, Brazil. [15]Khoury College of Computer Sciences, Northeastern University, Boston, MA 02115, USA. [16]Department of Bioengineering, School of Engineering and Applied Science, University of Pennsylvania, Philadelphia, PA 19104, USA. [17]Institute of Child Development, College of Education and Human Development, University of Minnesota, Minneapolis, MN 55455, USA. [18]Center for Biomedical Image Computing and Analytics, University of Pennsylvania, Philadelphia, PA 19104, USA. ✉e-mail: sattertt@pennmedicine.upenn.edu

