## [Peer Review File · Nature Communications]

Functional connectivity development along the sensorimotor-association axis enhances the cortical hierarchyEditorial Note: Parts of this Peer Review File have been redacted as indicated to remove third-party material where no permission to publish could be obtained.

REVIEWER COMMENTS

Reviewer #1 (Remarks to the Author):

This study uses four datasets to assess spatial patterns of functional brain development. Specifically, the study tests the hypothesis that the development of functional connectivity would map and diverge along the sensorimotor-association (S-A) axis. Results confirm the predictions, highlighting varied age-related changes in resting-state functional connectivity along the S-A axis.

This is a well-executed and written paper, with an emphasis on reproducibility. However, while the study provides some advances and confidence in current knowledge, such gain has a distinct incremental feeling. There is strong evidence, also generated by the group leading the current study (Sydnor et al. works), showing that cortical brain organisation and neurodevelopment map onto the S-A axis, with maturational patterns diverging mostly between the sensorimotor and association poles.

I also fear a level of redundancy among the various analyses. For example: (i) the positive age effect of somatomotor cortices and the negative effect in the association cortices presented in Figure 2 seem largely predictable from the results linked to Figure 1 and (ii) the finding that sensorimotor-to-sensorimotor edges tended to strengthen with age, whereas connections between sensorimotor and association regions weakened with development (Figure 7a-d) appears also predictable by FC patterns as a function of age described in Figure 1 (higher FC in somatomotor regions as a function of age and opposite trend in association cortices).

Functional connectivity based on correlation is a common but equivocal measure of coupling between brain regions. I appreciate the need to link to existing work but replicating the findings using measures like partial correlation (regressing out effects from all the other nodes in the network) may add confidence to the results.

Reviewer #2 (Remarks to the Author):

This manuscript examines the development of regional mean functional connectivity strength (FCS) in children and adolescents, showing that it mainly occurs along a sensorimotor-association (SA) axis of brain organization. A major strength of this study is that it examines four large samples, replicating the key findings across each through a pre-registered design. The main finding is that sensorimotor areas increase their FCS with age whereas association areas decrease their FCS. The analyses and main results are fairly straightforward, and the key point and has been demonstrated in some studies before. The major contribution here is to provide a robust demonstration of the key finding.

My main point is that the FCS maps in Figure 1 seem to contradict the existing literature. Multiple studies using fMRI and dMRI indicate that connectivity hubs of the brain reside in association areas (e.g., Buckner, et al. J Neurosci, 2009; Power et al. Neuron, 2012). This result is in line with common intuitions about how the brain is organized. Figure 1 suggests the opposite pattern with higher FCS in sensorimotor regions. This discrepancy needs to be explained before one can confidently interpret the findings, since the observation that the FCS of association areas decreases with age also appears to contradict multiple, diverse studies indicating that association areas develop the most during this time period, with the ongoing myelination of their connecting fibres being expected to increase FCS. A classic example is the development of frontal cortex, which this team has studied extensively. The statement on lines 382-384 seems to counter to this consensus view.

Given these considerations, I wonder whether the discrepant findings are driven the fact that FCS is computed as the mean of FC values obtained from data after global signal regression (GSR). The blend of negative and positive correlations that result from this procedure can make mean estimates such as FCS difficult to interpret, since an area with equally high levels of both negative

and positive FC would have FCS near zero. How do the results change if FCS is computed using the absolute correlation coefficient?

Another factor is the changing size of the networks to which each region belongs, which can affect estimates of FCS (Power et al. *Neuron*, 2012). This should also be considered.

Please explain precisely how a region's SA rank was defined.

I suggest reversing the colour bar in Fig 1 so that red represents positive values, which is more conventional in the field.

It would be useful to present data showing age effects on between-network and within-network connectivity separately for each network.

Figure 7 suggests there are three bands of age effects. Please provide a map showing which regions belong to each band.

Participants were excluded for motion if they showed mean FD above 0.3 mm. It is well-known (including from by this team) that subtle head motion can affect FC estimates. Some data on the number of frames exceeding common scrubbing criteria, and further analyses ruling out a contribution to FC estimates, would be helpful. The 10 mm threshold for frame exclusion is very liberal.

Some sections could be shortened considerably. The three sections spread across lines 167-312 simply show the same finding with different types of visualizations and laborious qualitative descriptions before presenting quantitative results. The sections are somewhat repetitive.

Reviewer #3 (Remarks to the Author):

The manuscript extends the authors' previously reported S-A axis principle in human brain organization (Syndor et al., *Neuron*, 2021) to the developmental trajectory in functional connectivity (FC). Through rigorous testing across four independent, large-scale neuroimaging datasets encompassing over 3000 participants, the authors explored the development of FC from childhood through adolescence. They observed increased functional connectivity in the sensorimotor cortex and a reduction in the association cortex during development. Overall, this is an important study with many strengths, including a pre-registered study design, replication across large sample sizes, multiple independent datasets enhancing reproducibility, and several sensitivity analyses concerning different parcellation schemes. There are several questions that should be addressed before publication.

1. The first concern is whether the decline in FC strength in the association cortex during development is biased by the negative, between-network FC observed in these areas. Figure 1 illustrates that the association cortex primarily exhibits negative connectivity, implying that a decrease in FC strength would result in an increase in absolute FC strength. Take the DMN as an example, its negative FC may be rationale, given its strong anticorrelation with attention networks. If this anticorrelation increases during development, then the FC may exhibit a more negative trend and would be interpreted as a "decline" in FC strength. However, this actually indicates a strengthened between-network relationship rather than functional segregations. I didn't mean to deny the existence of alterations in the DMN, but these aspects should be clarified to help readers better understand the findings.

2. In Figure 2, while the sensorimotor cortex demonstrates a consistent increment in FC, the primary visual cortex, being a part of the primary cortex and a lower-ranked area in the S-A axis, paradoxically exhibits a decrement in FC. This dissociation should be discussed.

3. Utilizing average FC with all other parcels as the primary outcome for each parcel may be influenced by global signal regression. The authors should assess and discuss the impact of global signal regression on the primary findings.

4. It would be helpful if the authors can present seed-based functional connectivity maps to demonstrate the developmental changes in FC. Selected parcels from M1, V1, DMN, and attention network could serve as the seeds for this analysis.

5. The results derived from the NKI dataset seem to diverge to some extent from those obtained from other datasets, as shown in Figure 2a and the curve pattern in Figure 5. A discussion exploring the potential factors contributing to these partial discrepancies is recommended.

Minor Issues:

1. In Figure 3, displaying the trajectories from all seven of Yeo's networks would be informative. Particularly, insights into the fronto-parietal network, which played a crucial role in maturity prediction in their prior work (Cui et al., *Neuron*, 2020), would be of interest.

2. Line 202: It should be clarified as "the predicted FC strength from GAMs" if the FC strength is indeed predicted from the GAMs.

3. Line 288: Are the mentioned five data points outliers? Displaying plots with all data points would be helpful.

4. Line 481: Please clarify what "third" refers to.

5. Line 755: It appears that some words might be missing.

6. Line 809: Please elaborate the methodology to generate the S-A ranks

Manuscript NCOMMS-23-40303-T:

Functional Connectivity Development along the Sensorimotor-Association Axis Enhances the Cortical Hierarchy

We thank the Editors of Nature Communications and the three reviewers of our manuscript for their time and for their valuable feedback. The constructive comments offered by reviewers have enhanced the clarity of our Results, Discussion, and Methods sections.

The reviewer's comments (*italicized*) and our responses are presented below. Modifications to the manuscript based on these comments are indicated in this response letter, as well as in the revised manuscript, *in blue text*.

REVIEWER #1

This study uses four datasets to assess spatial patterns of functional brain development. Specifically, the study tests the hypothesis that the development of functional connectivity would map and diverge along the sensorimotor-association (S-A) axis. Results confirm the predictions, highlighting varied age-related changes in resting-state functional connectivity along the S-A axis. This is a well-executed and written paper, with an emphasis on reproducibility.

We thank the reviewer for their positive comments on our work and these kind remarks.

1. However, while the study provides some advances and confidence in current knowledge, such gain has a distinct incremental feeling. There is strong evidence, also generated by the group leading the current study (Sydnor et al. works), showing that cortical brain organisation and neurodevelopment map onto the S-A axis, with maturational patterns diverging mostly between the sensorimotor and association poles.

We are happy to clarify the advances in the present study and distinctions from previous work. Sydnor et al. (2021)¹ was a review paper that proposed a hypothesis that the S-A axis represents not only an axis of cortical feature organization, but also an axis of cortical development. The review defined the S-A axis by averaging ten cortical maps capturing feature variation across the cortex. This review surveyed existing literature on brain organization and development and proposed a novel hypothesis that cortical development may be spatially and temporally organized by the S-A axis. While this hypothesis was supported by prior literature, this review and prior work did not empirically evaluate this hypothesis of hierarchical cortical development along the S-A axis.

Subsequently, Sydnor et al. (2023)² examined developmental refinements in intrinsic cortical activity, as quantified by the fluctuation amplitude of spontaneous BOLD activity, in a single dataset: the Philadelphia Neurodevelopmental Cohort. This study found that reductions in intrinsic activity amplitude unfolded heterochronously along the S-A axis. Although both features are defined from functional MRI, intrinsic fluctuation amplitude is quite different from functional connectivity; the former captures local regional dynamics and is thought to be linked to cortical microcircuits and plasticity, while the latter captures inter-regional signal coupling and may reflect long-range cortical connectivity and communication. Our present study employed functional connectivity, which is the single most commonly used measure of functional brain organization in human translational neuroscience. Despite this – and in contrast to measures of brain structure – the field lacks a definitive account regarding how functional connectivity develops in childhood and adolescence. Our present study used pre-registered hypotheses that were motivated by prior work; we replicated our findings across *four* datasets.

Notably, Sydnor et al. (2023) investigated patterns of developmental timing whereas our present study characterized spatial patterns of development along the S-A axis. Our present study found that the developmental program of the cortex led to the strengthening of hierarchical cortical organization. In contrast, Sydnor et al. (2023) did not examine how developmental trajectories may impact adult cortical feature variability.

In sum, though the present manuscript is grounded in the same overarching developmental hypothesis put forth in Sydnor et al. (2021),¹ it differs from these previous works in the measure under study, the biological phenomenon it captures, and its approach to replication across multiple independent datasets. Together, these features of our current study mark a significant advance over prior work.

2. I also fear a level of redundancy among the various analyses. For example: (i) the positive age effect of somatomotor cortices and the negative effect in the association cortices presented in Figure 2 seem largely predictable from the results linked to Figure 1 and (ii) the finding that sensorimotor-to-sensorimotor edges tended to strengthen with age, whereas connections between sensorimotor and association regions weakened with development (Figure 7a-d) appears also predictable by FC patterns as a function of age described in Figure 1 (higher FC in somatomotor regions as a function of age and opposite trend in association cortices).

We appreciate the reviewer for their suggestions for improving our figure presentation. In line with the reviewer's first suggestion (i), we have moved Figure 1 (predicted FC strength at ages 8, 14, and 22) to the supplement as **Supplementary Figure 7**.

We are also happy to clarify the rationale for Figure 7 (which is now updated to **Figure 6**), which is not redundant. In contrast to the regional analyses of Figures 1-5, Figure 6 focuses on effects of development at the edge level. The regional analyses of Figure 1-5 collapse information across edges, and as such cannot show how specific functional connections evolve with age. They depict the maturation of overall connectivity of a given region. For example, while regional analyses showed that sensorimotor regions tended to increase total FC strength, they did not reveal which specific connections drove that overall result. One plausible reason for a positive FC strength age effect for sensorimotor regions could be that sensorimotor regions maintained stable connectivity with other sensorimotor regions while strengthening connectivity to regions in the middle of the S-A axis. In contrast to this theoretical possibility, the edge-level analyses in Figure 6 provided novel information that sensorimotor-to-sensorimotor connections strengthened whereas sensorimotor-to-association connections weakened with age in development.

3. Functional connectivity based on correlation is a common but equivocal measure of coupling between brain regions. I appreciate the need to link to existing work but replicating the findings using measures like partial correlation (regressing out effects from all the other nodes in the network) may add confidence to the results.

We appreciate the reviewer's astute commentary on the limitations of functional connectivity as computed from Pearson correlations. As the reviewer notes, using measures like partial correlation may account for effects from other brain regions. However, Pearson correlation is currently by far the most commonly used measure to characterize functional connectivity in the field. Furthermore, we pre-registered functional connectivity strength as computed from standard Pearson correlations from XCP-D as our primary outcome.³ Additionally, partial correlation may not be appropriate for our primary analyses, which were focused on overall functional connectivity strength. Given that functional connectivity strength is a global measure that summarizes the connectivity between one region to all other regions, regressing out the effects of other nodes would change our pre-registered measure of interest.

While introducing a new measure of functional connectivity across four datasets would be beyond the scope of the current study, we have revised our Discussion to note this important limitation:

Lastly, while Pearson correlation is conventionally used to define functional connectivity,⁷⁶ this similarity measure is limited by factors such as nonlinear relationships and interactions between BOLD signals. Future work might also evaluate additional measures of functional connectivity, such as partial correlation and wavelet coherence.⁷⁶

REVIEWER #2

This manuscript examines the development of regional mean functional connectivity strength (FCS) in children and adolescents, showing that it mainly occurs along a sensorimotor-association (SA) axis of brain organization. A major strength of this study is that it examines four large samples, replicating the key findings across each through a pre-registered design. The main finding is that sensorimotor areas increase their FCS with age whereas association areas decrease their FCS. The analyses and main results are fairly straightforward, and the key point and has been demonstrated in some studies before. The major contribution here is to provide a robust demonstration of the key finding.

We appreciate the reviewer highlighting the strengths of this study.

1. My main point is that the FCS maps in Figure 1 seem to contradict the existing literature. Multiple studies using fMRI and dMRI indicate that connectivity hubs of the brain reside in association areas (e.g., Buckner, et al. J Neurosci, 2009; Power et al. Neuron, 2012). This result is in line with common intuitions about how the brain is organized. Figure 1 suggests the opposite pattern with higher FCS in sensorimotor regions. This discrepancy needs to be explained before one can confidently interpret the findings, since the observation that the FCS of association areas decreases with age also appears to contradict multiple, diverse studies indicating that association areas develop the most during this time period, with the ongoing myelination of their connecting fibres being expected to increase FCS. A classic example is the development of frontal cortex, which this team has studied extensively. The statement on lines 382-384 seems to counter to this consensus view.

This is excellent feedback, and we are happy to clarify. While Buckner, et al. J Neurosci, 2009⁴ and Power et al. Neuron, 2011⁵ are pivotal papers in the field, several important distinctions should be drawn between our study and these previous efforts. Furthermore, as detailed below, we believe that our additional analyses reveal that our findings are not contradictory with these prior results.

First, it is important to disambiguate both the goals and the methods of these prior papers compared to our study. These two papers aimed to characterize “hubness.” We would like to emphasize that this is not the goal of our work, which was focused on the development of functional connectivity. To study hubs, these two studies employed different methods and outcome measures. Specifically, Buckner et al. examined hubs as defined by degree centrality (the number of voxels across the brain that showed strong correlation with the voxel of interest) and betweenness centrality (the proportion of all shortest paths between pairs of regions that

included the region under study). Notably, Buckner et al. thresholded functional connectivity values at a high level ($r > 0.25$), which had the result of focusing the analysis only on the most strongly connected edges; all weakly connected edges and negative correlations were removed. This approach of defining connectivity is markedly different from our pre-registered strategy. We computed mean weighted edge strength and did not threshold matrices in our main analyses, allowing both weaker positive correlations as well as negative correlations to be examined. As described in our response to the next comments, we now include sensitivity analyses with thresholded matrices (excluding negative correlations).

Similarly, Power et al. also sought to define cortical hubs. However, this team used yet another graph-based measure to define hubs – the participation coefficient. Participation coefficient measures the extent to which a region connects to multiple other subnetworks (e.g., network communities) other than its own. Again, we would like to underscore that both the goals and analytic choices of these two (classic) studies are quite distinct from our pre-registered analyses focused on development.

Nonetheless, it should be noted that several of our findings do align with the network organization described by Power et al. Specifically, Power et al. reported that the default mode network (DMN) behaved like a processing system, with low participation coefficient and high system segregation. In contrast, the fronto-parietal network (FPN) displayed more diverse connections and had a higher participation coefficient. In our new supplementary analyses examining developmental changes in connectivity for specific networks (described in further detail in response to comment 6), we found that DMN segregated from other networks, reducing in connectivity to other networks with age. In contrast, the FPN increased in connectivity to salience / ventral attention, somatomotor, and visual networks with age. This analysis is displayed below as **Supplementary Figure 8**. Notably, Power and Buckner disagreed on how to understand the DMN. While Buckner interpreted the DMN as a prominent hub, Power et al. described DMN as “isolated” in the context of limited connectivity to other functional networks. The latter interpretation is more consistent with our results regarding the developmental segregation of the DMN.

Supplementary Figure 8. Developmental changes in connectivity between specific pairs of networks vary in direction and magnitude. The age effect of average connectivity between each pair of networks, derived from the Yeo 7 network solution, is displayed in each colored box. Within-network connectivity in visual, somatomotor, limbic, and fronto-parietal networks consistently increases with age. Connectivity within the default mode network also increases in HBN. Sensorimotor networks with the lowest average S-A ranks generally strengthen in between-network connectivity. The visual network shows most prominent increases with somatomotor, limbic, and fronto-parietal networks. The somatomotor network displays similar increases with age. In the higher order networks, limbic and fronto-parietal networks strengthen in between-network connectivity with all other networks except for default and dorsal attention networks. The highest ranked network – the default mode – tends to segregate from all other networks with age. Colored boxes represent the age effect of connectivity between each pair of networks, as computed by change in adjusted R^2 . FDR-corrected q-values: *** $q < 0.0001$, ** $q < 0.001$, * $q < 0.01$.

0.001, * $q < 0.05$. Pairs of networks that do not display significant change in connectivity over development are colored in white ($Q_{FDR} > 0.05$).

We have updated the Results section (“Hierarchical developmental changes in segregation and integration” subsection) accordingly:

Specifically, the lowest ranking networks, somatomotor and visual networks, tended to increase in connectivity to other networks, whereas the highest ranked network, DMN, tended to segregate from all other networks with age (**Supplementary Figures 8-10**).

And updated Methods (“Quantification of functional connectivity metrics” subsection) with:

Furthermore, to examine the development of connectivity between specific pairs of functional networks, we computed the average connectivity between each pair of networks as well as within-network connectivity (derived from Yeo’s 7-network solution). Of note, the average and range of S-A ranks for each network are as follows: visual (rank = 33, range = 2-109), somatomotor (rank = 40, range = 1-94), dorsal attention (rank = 84, range = 42-139), salience / ventral attention (rank = 113, range = 70-175), limbic (rank = 136, range = 108-167), fronto-parietal (rank = 144, range = 68-197), and default (rank = 155, range = 81-200).

Connectivity hubs tend to reside in association cortex, as the reviewer notes. However, hubs of connectivity to diverse parts of cortex are not found across all of association cortex, but are found specifically in heteromodal association cortex, which is thought to be responsible for executive function and cognitive control.⁶⁻⁸ This point is further supported by our new seed-based analysis examining age-related connectivity change in an exemplar region from the fronto-parietal control network (FPN) (**Supplementary Figure 14**). In addition to increasing connectivity to other FPN regions, this exemplar FPN region tended to increase in connectivity to diverse regions across multiple networks. In contrast, our seed-based analysis in DMN showed developmental decreases in connectivity to all other networks while specifically increasing connectivity to other DMN regions (**Supplementary Figure 15**). These findings are also consistent with Power et al.

Supplementary Figure 14. Maps of developmental change in connectivity for a seed region from the fronto-parietal control network (FPN). Maps are displayed for (a) PNC, (b) NKI, (c) HCP-D, and (d) HBN. This FPN region (red) tends to increase in within-network connectivity and moderately increase in connectivity to diverse regions in the visual, salience / ventral attention, and dorsal attention networks. All regions outlined in black display significant changes in connectivity ($Q_{\text{FDR}} < 0.05$). Results are shown in the Schaefer 200 atlas with a 7-network partition based on the 7 Yeo network solution.

Supplementary Figure 15. Age effect maps of connectivity for a seed region from the default mode network. Maps are displayed for (a) PNC, (b) NKI, (c) HCP-D, and (d) HBN. The default mode network region (red) increases in within-network connectivity while segregating from sensorimotor, salience / ventral attention, dorsal attention, and fronto-parietal control regions. Regions outlined in black exhibit significant changes in connectivity ($Q_{FDR} < 0.05$). Results are shown in the Schaefer 200 atlas with a 7-network partition based on the 7 Yeo network solution.

Furthermore, the reviewer notes that higher functional connectivity strength is seen in sensorimotor regions. We would like to clarify that the strong age effects of functional connectivity strength seen in somatomotor regions do *not* suggest that these regions are becoming more “hub-like” and connect to many diverse parts of cortex. In **Supplementary Figure 12**, we conducted a seed-based analysis of FC strength developmental change in the somatomotor network. The exemplar region from the somatomotor network displayed strong increases in connectivity specifically to other sensorimotor regions. Because functional connectivity strength was computed from the average of each region’s overall connectivity, the effects seen in functional connectivity strength were driven by these strong local changes in connectivity. These results are consistent with prior work that reported high levels of local connectivity in sensorimotor regions.⁹ Hence, our developmental findings do not imply the sensorimotor regions are hubs in the classical sense of having strong connectivity to many other cortical networks or communities.

Supplementary Figure 12. Maps of developmental change in connectivity for a seed region from the somatomotor network. Spatial patterns for the somatomotor seed region are generally consistent across (a) PNC, (b) NKI, (c) HCP-D, and (d) HBN. The somatomotor region (red) tends to increase in connectivity to other somatomotor and some visual regions, while

segregating from regions within the DMN. Regions outlined in black exhibit significant changes in connectivity ($Q_{FDR} < 0.05$). Results are shown in the Schaefer 200 atlas with a 7-network partition based on the 7 Yeo network solution.

These seed-based analyses were included in the revised Results (“Hierarchical developmental changes in segregation and integration” subsection) as follows:

Seed-based analyses using edge-level data examined age effects from exemplar regions from the visual, somatomotor, salience / ventral attention, fronto-parietal, and default mode networks... Somatomotor regions tended to increase in connectivity with other sensorimotor regions while decreasing in connectivity with some regions in the DMN (**Supplementary Figures 11**)... In addition to increasing connectivity to other fronto-parietal network regions, the fronto-parietal seed tended to increase in connectivity to diverse regions across multiple networks (**Supplementary Figure 14**). The DMN seed showed developmental decreases in connectivity to all other networks while increasing connectivity to other DMN regions (**Supplementary Figure 15**).

And were included in the revised Methods section (“Quantification of functional connectivity metrics” subsection) as follows:

Seed-based analyses examining age-related connectivity in exemplar regions from the visual, somatomotor, salience / ventral attention, fronto-parietal control, and default mode networks were conducted using edge-level connectivity data.

Lastly, we agree with the reviewer that frontal regions are expected to undergo protracted myelination into adolescence and young adulthood. While functional and structural connectivity (derived from fMRI and dMRI, respectively) often align, they are thought to index different properties, with fMRI measuring polysynaptic connections that cannot be measured using dMRI. Furthermore, the correspondence between structure and function using dMRI and fMRI is far from a direct relationship,¹⁰ especially in higher-order regions.^{11,12} In higher-order association regions, functional connectivity is thought to be less constrained by direct structural links and rely more on indirect, polysynaptic connections.

2. Given these considerations, I wonder whether the discrepant findings are driven by the fact that FCS is computed as the mean of FC values obtained from data after global signal regression (GSR). The blend of negative and positive correlations that result from this procedure can make mean estimates such as FCS difficult to interpret, since an area with equally high levels of both negative and positive FC would have FCS near zero. How do the results change if FCS is computed using the absolute correlation coefficient?

We agree with the reviewer that it would be useful to clarify the impact of anti-correlations in the context of preprocessing with GSR. To evaluate this according to the reviewer's suggestions, we conducted three new supplementary analyses using data from the PNC. In the first analysis, as suggested, we used the absolute value of the correlation coefficient as the measure of functional connectivity following preprocessing with GSR. In the second analysis, we again preprocessed the data using GSR but then thresholded the connectivity matrices to include only positive correlations. In our third analysis, we preprocessed the data without GSR and retained signed edges without thresholding. FC strength was then computed on connectivity matrices that were derived from this data.

Notably, all three analyses yielded convergent results: the development of FC strength aligned with the sensorimotor-association axis. When negative connectivity values were removed by either evaluating the absolute value of FC (analysis 1; GSR+) or thresholding the connectivity matrices at zero (analysis 2; GSR+), we found that age effects were still significantly aligned with the S-A axis. However, we did find that in association regions at the top of the S-A axis (primarily in the default network), negative age effects were largely abolished as most effects were positive. These analyses are plotted below in **Supplementary Figure 6b and c**. Finally, in our analyses that excluded GSR (analysis 3; GSR -; **Supp. Figure 6d**), age effects that were negative in association regions in the original analyses remained negative, supporting that the negative age effects present in original analyses in fact were not due to GSR.

Supplementary Figure 6. Sensitivity analyses using absolute correlation, thresholded connectivity matrices, and no global signal regression to compute functional connectivity strength provide convergent results. (a) Original analysis from the PNC correlating the age effect of functional connectivity (FC) strength with sensorimotor-association axis rank is displayed for comparison to results of additional analyses. Sensitivity analyses in the PNC were conducted using FC strength computed (b) using the absolute value of the correlation coefficient as the measure of functional connectivity, (c) from thresholded connectivity matrices (including positive correlations only), and (d) without global signal regression. Consistent with our main findings using global signal regression and Pearson correlation, the development of FC strength varies continuously along the S-A axis. (b-d) The rank of each region in the S-A axis is significantly associated with FC strength age effects in the PNC using absolute correlation ($r = -0.48, p_{spin} < 0.001$), thresholded matrices ($r = -0.49, p_{spin} < 0.001$), and no global signal regression ($r = -0.59, p_{spin} < 0.0001$). In association regions at the top of the S-A axis (primarily in the default network) in (b) and (c), negative age effects are abolished and are found to be positive. The age effect of FC strength remains negatively associated with S-A axis rank even after excluding GSR (d), suggesting that GSR played a minimal role in our findings.

We have added the following to our Results section (“Development of functional connectivity varies along the S-A axis” subsection):

Results remained consistent after additional sensitivity analyses, which computed FC strength in three additional ways: using the absolute value of the correlation coefficient as the measure of functional connectivity, thresholding connectivity matrices to only include positive correlations, and excluding global signal regression during preprocessing (**Supplementary Figure 6**).

And updated our Methods (“Sensitivity Analysis” subsection) to include these additional analyses:

To evaluate whether the main findings with FC strength were impacted by negative correlations and global signal regression (GSR), FC strength was computed in three additional ways. First, the absolute value of the correlation coefficient was used as the measure of functional connectivity following preprocessing with GSR. Second, after processing the data using GSR, connectivity matrices were thresholded to include only positive correlations. FC strength was then computed on thresholded matrices. Third, the data was processed without GSR, and signed edges were retained without thresholding.

To further investigate how FC strength changed across development in association regions and extend the findings presented in **Supplementary Figure 6**, we plotted the predicted developmental trajectories for highly transmodal regions at the top of the S-A axis (e.g., regions ranking 180 or higher on the S-A axis; most of these regions were part of the DMN). Note that this analysis is similar to that of Figure 3 (GSR+; using original, signed connectivity matrices), except that smooth fits were not zero centered. A majority of these association regions displayed positive FC strength in childhood that became negative through adolescence (producing negative age effects in the original analysis). Several association regions (almost all within the DMN) displayed negative FC strength in childhood that became increasingly negative with development (again, producing negative age effects in the original analysis). This figure is displayed below as **Supplementary Figure 4**. This analysis helps explain the results of our sensitivity analysis using the absolute value (analysis 1; GSR+): regions that exhibited increasingly negative FC strengths with age tended to display positively-signed age effects when an absolute value was applied.

Supplementary Figure 4. Functional connectivity strength decreases in association regions with age. Developmental trajectories for predicted functional connectivity strength in the highest ranking regions (rank > 180; most are part of default mode network) are shown for (a) PNC, (b) NKI, (c) HCP-D, and (d) HBN. Regions in the association pole show decreases in functional connectivity strength through development. A majority of regions display positive functional connectivity strength in childhood that becomes negative through adolescence. Several regions display increasingly negative functional connectivity strength with age; most of these regions were part of the default mode network. Each line represents the functional connectivity strength for each region throughout development, modeled using generalized additive models.

We have modified our Results section (“Development of functional connectivity varies along the S-A axis” subsection) to include this analysis:

Brain regions in this network decreased in FC strength through development (**Figure 2m-p** and **Supplementary Figure 4**).

A related effect occurred when matrices were thresholded to retain only positive connections (analysis 2; GSR+): among the highest ranking regions on the S-A axis (again, rank > 180) that switched from negative to positive age effects using thresholded matrices, 75% were members of the DMN. Among the DMN regions that switched from negative to positive age effects, 50% of the connections that remained following thresholding were to other DMN regions. Connections to regions outside the DMN, which tended to be negative, were removed in thresholding. This indicates that increasing within-network connectivity for DMN regions likely drove the observed positive age effects seen for regions at the top of the S-A axis when thresholded matrices were used in analysis 2. This analysis supports that decreases in FC strength seen in association areas in our original analyses were due to increasingly negative between-network DMN correlations.

Lastly, our analysis that excluded GSR (analysis 3; GSR -) confirms that our original findings of negative age effects in association regions were not driven by GSR. Consistent with original findings, FC strength increased in sensorimotor regions with age whereas FC strength in association cortices tended to decrease with age (**Supplementary Figure 6d**). Furthermore, spatial maps of mean FC strength across the cortex with and without GSR both showed that somatomotor regions tended to have higher FC strength whereas regions in the default mode network tended to have lower FC strength.. Both spatial maps were significantly correlated with S-A axis rank (with GSR: $r = -0.48$, $p_{\text{spin}} < 0.0001$; without GSR: $r = -0.45$, $p_{\text{spin}} < 0.0001$). Taken together, the age effect of FC strength remains negatively associated with S-A axis rank even after excluding GSR, suggesting that GSR played a minimal role in our findings.

3. Another factor is the changing size of the networks to which each region belongs, which can affect estimates of FCS (Power et al. Neuron, 2012). This should also be considered.

We conducted extensive supplementary analyses that used multiple cortical atlases and network partitions. Across these analyses, the size of the regions themselves and the size of the network partitions to which regions are assigned vary considerably. As depicted in **Supplementary Figure 2** (below) results were remarkably similar in all cases, bolstering confidence in our findings and emphasizing that variation in node or network size did not drive our results.

Supplementary Figure 2. Sensitivity analysis with secondary parcellations to examine functional connectivity strength development provides convergent results. We computed functional connectivity strength from concatenated resting-state and task fMRI for additional parcellations: HCP-MMP, Gordon, and Schaefer 400 atlases.^{1,4,5} Representative results from sensitivity analyses in the PNC are depicted here. **(a-c)** Similar to our main findings using the primary parcellation (Schaefer 200 atlas), dissociable patterns of functional connectivity strength developmental trajectories can be seen along the S-A axis across all secondary parcellations. **(d-f)** The strong alignment between the age effect of functional connectivity strength and a given region’s rank on the S-A axis is consistent across parcellations (HCP-MMP: $r = -0.72$, $p_{\text{spin}} < 0.0001$; Gordon: $r = -0.71$, $p_{\text{spin}} < 0.0001$; Schaefer 400: $r = -0.70$, $p_{\text{spin}} < 0.0001$).

4. Please explain precisely how a region’s SA rank was defined.

We are happy to clarify. We have revised our Methods (“Alignment of functional connectivity metric age effects to the S-A axis” subsection) to further detail how a region’s S-A axis rank was defined:

The S-A axis was previously derived in Sydnor et al. (2021)¹⁹ from multi-modal brain maps that exhibit stereotyped feature variability between primary sensorimotor and transmodal association cortices. As a result, the S-A axis represents the average cortical

hierarchy derived from multivariate brain properties. A vertex-level S-A axis (in the fsLR surface-density 32k) was parcellated with study atlases to yield regional S-A axis ranks. The S-A axis is publicly available and was obtained from https://github.com/PennLINC/S-A_ArchetypalAxis for this study.

Specifically, ten cortical maps were used to derive the S-A axis. These ten maps included anatomical hierarchy as quantified by the T1-weighted to T2-weighted ratio,¹⁰⁴ functional hierarchy,⁵ evolutionary hierarchy,¹⁰⁵ allometric scaling,¹⁰⁶ aerobic glycolysis,¹⁰⁷ cerebral blood flow,¹⁰⁸ gene expression,¹⁰⁹ first principal component of NeuroSynth terms,¹¹⁰ externopyramidization,¹¹¹ and cortical thickness.¹⁹ Vertex-level representations for each of the ten maps were obtained. Vertices were rank-ordered based on feature value. Vertex rankings were then averaged across all ten maps to derive the spatial patterning of feature variability along a unidimensional S-A axis. The vertex-level S-A axis was parcellated into five cortical atlases used in this paper (Schaefer 200, Schaefer 400, HCP-MMP, and Gordon) such that each region was assigned a unique S-A rank indicating its relative position on the S-A axis.

5. I suggest reversing the colour bar in Fig 1 so that red represents positive values, which is more conventional in the field.

We appreciate the reviewer's suggestion. As suggested, we have changed the color such that orange indicates positive FC strength and teal indicates negative FC strength. Furthermore, to streamline the presentation of our results in the main text, we have moved Figure 1 to the supplementary information to become the new **Supplementary Figure 7**.

Supplementary Figure 7. The spatial distribution of functional connectivity strength is highly similar across all four datasets and is refined with age. The refinement of functional connectivity (FC) strength across the cortex appears highly similar across all four datasets. In the Philadelphia Neurodevelopmental Cohort (PNC), (a) the fitted values of FC strength predicted from regional GAMs are highest in somatomotor cortices in childhood and increase in these cortical areas with age. In contrast, fitted FC strength in association cortices is lower in childhood and tends to decrease with age. This decrease results in negative FC strength values in transmodal association cortices by early adulthood, suggestive of weakly anti-correlated connectivity with most brain regions. Similar spatial patterns of FC strength are seen in (b) Nathan Kline Institute-Rockland (NKI), (c) Human Connectome Project Development (HCP-D), and (d) Healthy Brain Network (HBN). In a-d, the predicted value of FC strength in each region is shown at ages 8, 14, and 22 across the four datasets. Generalized additive models (GAM) were fitted independently for each cortical region and used to predict the fitted value of FC

strength at each age. Each row of plots corresponds to results from a given dataset. Columns represent the FC strength map at each age. Results are shown in the Schaefer 200 atlas.

6. It would be useful to present data showing age effects on between-network and within-network connectivity separately for each network.

We thank the reviewer for this helpful comment. In line with the reviewer’s suggestions, we have added this analysis to our revised manuscript, copied below as the new **Supplementary Figure 8**.

Supplementary Figure 8. Developmental changes in connectivity between specific pairs of networks vary in direction and magnitude. The age effect of average connectivity between each pair of networks, derived from the Yeo 7 network solution, is displayed in each colored box. Within-network connectivity in visual, somatomotor, limbic, and fronto-parietal networks

consistently increases with age. Connectivity within the default mode network also increases in HBN. Sensorimotor networks with the lowest average S-A ranks generally strengthen in between-network connectivity. The visual network shows most prominent increases with somatomotor, limbic, and fronto-parietal networks. The somatomotor network displays similar increases with age. In the higher order networks, limbic and fronto-parietal networks strengthen in between-network connectivity with all other networks except for default and dorsal attention networks. The highest ranked network – default mode – tends to segregate from all other networks with age. Colored boxes represent the age effect of connectivity between each pair of networks, as computed by change in adjusted R^2 . FDR-corrected q-values: *** $q < 0.0001$, ** $q < 0.001$, * $q < 0.05$. Pairs of networks that do not display significant change in connectivity over development are colored in white ($Q_{FDR} > 0.05$).

We have updated the Results section (“Hierarchical developmental changes in segregation and integration” subsection):

Specifically, the lowest ranking networks, somatomotor and visual networks, tended to increase in connectivity to other networks, whereas the highest ranked network, DMN, tended to segregate from all other networks with age (**Supplementary Figures 8-10**).

And we have updated Methods accordingly (“Quantification of functional connectivity metrics” subsection):

Furthermore, to examine the development of connectivity between specific pairs of functional networks, we computed the average connectivity between each pair of networks as well as within-network connectivity (derived from Yeo’s 7-network solution). Of note, the average and range of S-A ranks for each network are as follows: visual (rank = 33, range = 2-109), somatomotor (rank = 40, range = 1-94), dorsal attention (rank = 84, range = 42-139), salience / ventral attention (rank = 113, range = 70-175), limbic (rank = 136, range = 108-167), fronto-parietal (rank = 144, range = 68-197), and default (rank = 155, range = 81-200).

7. Figure 7 suggests there are three bands of age effects. Please provide a map showing which regions belong to each band.

We are happy to clarify. In contrast to the other analyses in our paper that focus on brain regions, this analysis is focused on edges (or functional connections). Specifically, Figure 7 (now updated to the new **Figure 6**) displays the age effect for the functional connections between each pair of brain regions. For each plot, the bands that the reviewer refers to correspond to sensorimotor-to-sensorimotor connections (bottom left: yellow, indicating strong increases in

connectivity), connections between middle-axis regions (middle: white or light purple, indicating lack of change or modest decreases in connectivity), and sensorimotor-to-association connections (top left and bottom right: dark purple; strong decreases in connectivity). Furthermore, the top right represents association-to-association connections, which are light purple, suggesting modest decreases in connectivity with age. Because this figure displays the age effect of network edges rather than of regions, this data cannot be displayed on a cortical map. In an effort to clarify, we have revised the caption for this figure.

Figure 6. Edge-level age effects confirm divergent connectivity refinement along the sensorimotor-association axis. (a-d) Topographical plots display edge-level connectivity age effects as a function of S-A axis rank. Colors indicate the magnitude and direction of the age effect at each region-to-region connection. Contour lines indicate a constant age effect. Sensorimotor-to-sensorimotor connections (bottom left of each plot; yellow) display strong increases in connectivity. Connections between middle-axis regions (middle of the figure; white or light purple) demonstrate lack of change or modest decreases in connectivity. Sensorimotor-to-association connections (top left and bottom right; dark purple) show strong decreases in connectivity. Association-to-association connections (top right; light purple) also moderately weaken in development. Note that plots display the age effect of edges rather than of regions.

8. Participants were excluded for motion if they showed mean FD above 0.3 mm. It is well-known (including from by this team) that subtle head motion can affect FC estimates. Some data on the number of frames exceeding common scrubbing criteria, and further analyses ruling out a contribution to FC estimates, would be helpful. The 10 mm threshold for frame exclusion is very liberal.

We handled motion in three ways. First, in our study's inclusion criteria, we applied a threshold of 0.3 mm; this is a strict and widely-used threshold in the field.^{13,14} Second, we included mean FD as a covariate in all analyses, controlling for any residual impact of head motion even following denoising and this strict inclusion criteria. Third and finally, we used a top-performing denoising procedure that was selected on the basis of our prior benchmarking studies,¹⁵ which have been replicated by independent groups.¹⁶ This previous work showed that the pipeline used in this paper – which includes 36 confound signals as well as despiking (36+despike in figure below) – performed as well as a similar pipeline that included scrubbing (36+scrub in figure below).

[REDACTED]

We are happy to clarify about the 10 mm threshold; this was poorly worded. The 10 mm threshold for frame exclusion that the reviewer refers to was applied at this step with the purpose of effectively turning off scrubbing at the preprocessing step. We have revised the statement in the Methods to make this clear (below).

Then, the BOLD data were despiked, mean-centered, and linearly detrended. Despiking is a temporal censoring operation that performs similarly to scrubbing in prior benchmarking studies.⁷⁵

9. Some sections could be shortened considerably. The three sections spread across lines 167-312 simply show the same finding with different types of visualizations and labor various qualitative descriptions before presenting quantitative results. The sections are somewhat repetitive.

We thank the reviewer for this suggestion. As suggested, we have moved Figure 1 to the supplement (new **Supplementary Figure 7**) to streamline our Results section.

REVIEWER #3

The manuscript extends the authors' previously reported S-A axis principle in human brain organization (Syndor et al., Neuron, 2021) to the developmental trajectory in functional connectivity (FC). Through rigorous testing across four independent, large-scale neuroimaging datasets encompassing over 3000 participants, the authors explored the development of FC from childhood through adolescence. They observed increased functional connectivity in the sensorimotor cortex and a reduction in the association cortex during development.

Overall, this is an important study with many strengths, including a pre-registered study design, replication across large sample sizes, multiple independent datasets enhancing reproducibility, and several sensitivity analyses concerning different parcellation schemes.

We thank the reviewer for their positive appraisal of our work.

There are several questions that should be addressed before publication.

1. The first concern is whether the decline in FC strength in the association cortex during development is biased by the negative, between-network FC observed in these areas. Figure 1 illustrates that the association cortex primarily exhibits negative connectivity, implying that a decrease in FC strength would result in an increase in absolute FC strength. Take the DMN as an example, its negative FC may be rationale, given its strong anticorrelation with attention networks. If this anticorrelation increases during development, then the FC may exhibit a more negative trend and would be interpreted as a "decline" in FC strength. However, this actually

indicates a strengthened between-network relationship rather than functional segregations. I didn't mean to deny the existence of alterations in the DMN, but these aspects should be clarified to help readers better understand the findings.

We are happy to clarify as these are important points. First, to investigate the role of negatively-signed connections, we conducted three new sensitivity analyses in the PNC. In the first analysis, we used the absolute value of the correlation coefficient as the measure of functional connectivity following preprocessing with GSR; this yielded only positively-signed connections. In the second analysis, we conducted an analysis where we thresholded connectivity matrices at zero, removing any negatively-signed connections. In our third analysis, we preprocessed the data without GSR and retained signed edges without thresholding.

All three analyses yielded convergent results: the development of FC strength aligned with the sensorimotor-association axis. When negative connectivity values were removed by either evaluating the absolute value of FC (analysis 1; GSR+) or thresholding the connectivity matrices at zero (analysis 2; GSR+), we found that age effects were still significantly aligned with the S-A axis. However, we did find that in association regions at the top of the S-A axis (primarily in the default network), negative age effects were abolished and were found to be positive. These analyses are plotted below in **Supplementary Figure 6b and c**. Finally, in our analyses that excluded GSR (analysis 3; GSR -; **Supp. Figure 6d**), age effects that were negative in association regions in the original analyses remained negative, supporting that the negative age effects present in original analyses in fact were not due to GSR.

Supplementary Figure 6. Sensitivity analyses using absolute correlation, thresholded connectivity matrices, and no global signal regression to compute functional connectivity strength provide convergent results. (a) Original analysis from the PNC correlating the age effect of functional connectivity (FC) strength with sensorimotor-association axis rank is displayed for comparison to results of additional analyses. Sensitivity analyses in the PNC were conducted using FC strength computed (b) using the absolute value of the correlation coefficient as the measure of functional connectivity, (c) from thresholded connectivity matrices (including positive correlations only), and (d) without global signal regression. Consistent with our main findings using global signal regression and Pearson correlation, the development of FC strength varies continuously along the S-A axis. (b-d) The rank of each region in the S-A axis is significantly associated with FC strength age effects in the PNC using absolute correlation ($r = -0.48, p_{spin} < 0.001$), thresholded matrices ($r = -0.49, p_{spin} < 0.001$), and no global signal regression ($r = -0.59, p_{spin} < 0.0001$). In association regions at the top of the S-A axis (primarily in the default network) in (b) and (c), negative age effects are abolished and are found to be positive. The age effect of FC strength remains negatively associated with S-A axis rank even after excluding GSR (d), suggesting that GSR played a minimal role in our findings.

We have added the following to our Results section (“Development of functional connectivity varies along the S-A axis” subsection):

Results remained consistent after additional sensitivity analyses, which computed FC strength in three additional ways: using the absolute value of the correlation coefficient as the measure of functional connectivity, thresholding connectivity matrices to only include positive correlations, and excluding global signal regression during preprocessing (**Supplementary Figure 6**).

And updated our Methods (“Sensitivity Analysis” subsection) to include these additional analyses:

To evaluate whether the main findings with FC strength were impacted by negative correlations and global signal regression (GSR), FC strength was computed in three additional ways. First, the absolute value of the correlation coefficient was used as the measure of functional connectivity following preprocessing with GSR. Second, after processing the data using GSR, connectivity matrices were thresholded to include only positive correlations. FC strength was then computed on thresholded matrices. Thirdly, the data was processed without GSR, and signed edges were retained without thresholding. FC strength was then computed on connectivity matrices that were derived from this data.

To investigate this further, we plotted the predicted developmental trajectory for regions at the top of the S-A hierarchy (e.g., rank > 180; most of these regions were part of the DMN). In our original analyses that used signed values without thresholding, a majority of these association regions displayed positive FC strength in childhood that became negative through adolescence (producing negative age effects in the original analysis). Several association regions (almost all within the DMN) displayed negative FC strength in childhood that became increasingly negative with development (again, producing negative age effects in the original analysis). This figure supports the reviewer’s comments that the anti-correlation seen in some association regions (many of which are in the DMN) increases during development. This figure is displayed below and is now included in the manuscript as a new **Supplementary Figure 4**.

Supplementary Figure 4. Functional connectivity strength decreases in association regions with age. Developmental trajectories for predicted functional connectivity strength in the highest ranking regions (rank > 180; most are part of default mode network) are shown for (a) PNC, (b) NKI, (c) HCP-D, and (d) HBN. Regions in the association pole show decreases in functional connectivity strength through development. A majority of regions display positive functional connectivity strength in childhood that becomes negative through adolescence. Several regions display increasingly negative functional connectivity strength with age; most of these regions were part of the default mode network. Each line represents the functional connectivity strength for each region throughout development, modeled using generalized additive models.

We have modified our Results section (“Development of functional connectivity varies along the S-A axis” subsection) to include this analysis:

Brain regions in this network decreased in FC strength through development (**Figure 2m-p and Supplementary Figure 4**).

We next examined between-network connectivity developmental changes specifically for the DMN, which includes numerous regions with negative connectivity values. New **Supplementary Figure 10** shows predicted developmental trajectories for the mean connectivity of DMN to each of Yeo’s 7 networks. **Supplementary Figure 10a** displays developmental trajectories for connectivity between DMN and other networks; these are zero-centered in **Supplementary Figure 10b** to better visualize directional effects. Our findings revealed that DMN connectivity with most networks (fronto-parietal, salience / ventral attention, somatomotor, visual, and limbic) became increasingly negative with age. This suggests that the DMN became increasingly segregated from these other networks with development.

Supplementary Figure 10. Developmental changes in between-network connectivity for default mode network. (a-b) In PNC, NKI, HCP-D, and HBN, default mode network (DMN) decreases in mean connectivity to all other networks with age. DMN connectivity to fronto-parietal, salience / ventral attention, somatomotor, visual, and limbic networks tends to be negative or close to zero in childhood and becomes increasingly negative with development.

(a) Predicted mean connectivity generated from GAMs fit on each pair of networks are displayed for each dataset. (b) Zero-centered developmental trajectories that display directional changes are plotted for each dataset.

We have revised our Results section (“Hierarchical developmental changes in segregation and integration” subsection) to include this analysis:

...the highest ranked network, DMN, tended to segregate from all other networks with age (**Supplementary Figures 8-10**).

And updated our Methods section (“Quantification of functional connectivity metrics” subsection) accordingly:

Furthermore, to examine the development of connectivity between specific pairs of functional networks, we computed the average connectivity between each pair of networks as well as within-network connectivity (derived from Yeo’s 7-network solution).

2. In Figure 2, while the sensorimotor cortex demonstrates a consistent increment in FC, the primary visual cortex, being a part of the primary cortex and a lower-ranked area in the S-A axis, paradoxically exhibits a decrement in FC. This dissociation should be discussed.

We agree that this is an unexpected result. We further investigated the development of FC strength in the visual network and found that most of the significant age effects seen in visual regions were positive, as expected. Specifically in the primary visual cortex, the decrease in FC strength was significantly negative in three out of four datasets. We have added the following text to the Discussion to comment on this unexpected finding:

Of note, unlike the rest of the sensorimotor cortices, the primary visual cortex showed a unique decrease in functional connectivity strength during development. This finding may reflect the need for segregated visual processing in this region.

Furthermore, we would expect visual and somatomotor networks to exhibit similar developmental patterns. To investigate how visual and somatomotor networks may differ in maturational patterns, we examined the developmental changes in connectivity between each of these two networks with all other functional networks. Both networks showed increasingly negative connectivity to the DMN and dorsal attention network, but increased in connectivity to all other networks. Within-network connectivity also increased. Therefore, while V1 exhibited a

unique decrease in FC strength with development, we confirmed that the developmental changes in overall connectivity for visual and somatomotor networks were highly similar. This analysis is included as **Supplementary Figure 9**, displayed below.

Supplementary Figure 9. Comparing between- and within-network connectivity developmental trajectories in visual and somatomotor networks. For each dataset, the developmental change in mean connectivity of (a-b) visual and (c-d) somatomotor networks to all other networks is plotted. In (a) and (c), predicted mean connectivity generated from GAMs fit on each pair of networks are displayed for each dataset. In (b) and (d), zero-centered developmental trajectories that display directional changes are plotted for each dataset. Visual and somatomotor networks exhibit highly similar developmental changes in connectivity. Networks are partitioned using the Yeo 7 network solution.

We have modified the Results section (“Hierarchical developmental changes in segregation and integration” subsection) accordingly:

Specifically, the lowest ranking networks, somatomotor and visual networks, tended to increase in connectivity to other networks...with age (**Supplementary Figures 8-10**).

As well as the Methods section (“Quantification of functional connectivity metrics” subsection):

Furthermore, to examine the development of connectivity between specific pairs of functional networks, we computed the average connectivity between each pair of networks as well as within-network connectivity (derived from Yeo’s 7-network solution).

3. Utilizing average FC with all other regions as the primary outcome for each region may be influenced by global signal regression. The authors should assess and discuss the impact of global signal regression on the primary findings.

We are happy to address this. It is well-described that the use of GSR in preprocessing can induce anticorrelations in FC. Negatively-signed connectivity values may impact our primary measure of FC strength, which is the average of each region’s correlations with all other regions.

As the reviewer suggested, we conducted a sensitivity analysis that excluded GSR as a preprocessing step (**Supp. Figure 6d**, also described in detail in our response to the reviewer’s comment 1). We found that the relationship between age effect for FC strength development and sensorimotor-association axis rank remained strong and significant, suggesting that GSR did not impact our primary findings. This analysis was completed in PNC and is plotted in the new **Supplementary Figure 6d** copied below:

Supplementary Figure 6. ...Sensitivity analyses in the PNC were conducted using FC strength computed... (d) without global signal regression. Consistent with our main findings using global signal regression and Pearson correlation, the development of FC strength varies continuously along the S-A axis...The rank of each region in the S-A axis is significantly associated with FC strength age effects in the PNC using... no global signal regression ($r = -0.59, p_{spin} < 0.0001$).

The following text was added accordingly to the Methods section (“Sensitivity Analysis” subsection):

To evaluate whether the main findings with FC strength were impacted by negative correlations and global signal regression (GSR), FC strength was computed in... additional ways... The data was processed without GSR, and signed edges were retained without thresholding. FC strength was then computed on connectivity matrices that were derived from this data.

4. It would be helpful if the authors can present seed-based functional connectivity maps to demonstrate the developmental changes in FC. Selected regions from M1, V1, DMN, and attention network could serve as the seeds for this analysis.

This is a helpful suggestion. We now include multiple seed analyses as suggested. These seeds include regions within the visual, somatomotor, salience / ventral attention, and default mode networks. Each region’s age effect of connectivity to all other regions was computed by fitting edge-level GAMs (as previously described in the Methods). As shown in the newly added

Supplementary Figures 11, 12, 13, and 15 below, visual and somatomotor regions tended to increase most in connectivity with other sensorimotor regions while decreasing in connectivity with regions in the DMN. The seed within the salience / ventral attention network generally integrated with other attention network regions and with somatomotor regions, but decreased in connectivity with DMN. Lastly, consistent with previous literature,¹⁷ DMN regions increased in connectivity with other DMN regions while segregating from all other networks.

Supplementary Figure 11. Age effect maps of connectivity for a select region from the visual network. Maps are shown for (a) PNC, (b) NKI, (c) HCP-D, and (d) HBN. Age effects of connectivity between the indicated visual seed region (red) and the rest of cortex are shown on the cortical surface with pink indicating increasing connectivity with age and teal indicating decreasing connectivity with age. The visual seed region tends to increase most in connectivity to somatomotor and other visual regions while segregating from DMN regions. All regions outlined in black display significant changes in connectivity ($Q_{FDR} < 0.05$). Results are shown in the Schaefer 200 atlas with a 7-network partition based on the 7 Yeo network solution.

Supplementary Figure 12. Maps of developmental change in connectivity for a seed region from the somatomotor network. Spatial patterns for the somatomotor seed region are generally consistent across (a) PNC, (b) NKI, (c) HCP-D, and (d) HBN. The somatomotor region (red) tends to increase most in connectivity to other somatomotor regions and moderately to visual regions, while segregating from regions within the DMN. Regions outlined in black exhibit significant changes in connectivity ($Q_{FDR} < 0.05$). Results are shown in the Schaefer 200 atlas with a 7-network partition based on the 7 Yeo network solution.

Supplementary Figure 13. Age effect maps for connectivity in a select region from the salience / ventral attention network. Maps are depicted for (a) PNC, (b) NKI, (c) HCP-D, and (d) HBN. The region from the salience / ventral attention network (red) strengthens in connectivity with other salience / ventral attention network regions and with somatomotor regions, but decreases in connectivity with DMN. Regions outlined in black exhibit significant changes in connectivity ($Q_{FDR} < 0.05$). Results are shown in the Schaefer 200 atlas with a 7-network partition based on the 7 Yeo network solution.

Supplementary Figure 15. Age effect maps of connectivity for a seed region from the default mode network. Maps are displayed for (a) PNC, (b) NKI, (c) HCP-D, and (d) HBN. The default mode network region (red) increases in within-network connectivity while segregating from sensorimotor, salience / ventral attention, dorsal attention, and fronto-parietal control regions. Regions outlined in black exhibit significant changes in connectivity ($Q_{FDR} < 0.05$). Results are shown in the Schaefer 200 atlas with a 7-network partition based on the 7 Yeo network solution.

We updated the text in our Results section (“Hierarchical developmental changes in segregation and integration” subsection) to include the following:

Seed-based analyses using edge-level data examined age effects from exemplar regions from the visual, somatomotor, salience / ventral attention, fronto-parietal, and default mode network. Visual and somatomotor regions tended to increase most in connectivity with other sensorimotor regions while decreasing in connectivity with regions in the

DMN (**Supplementary Figures 11-12**). The seed from the salience / ventral attention network generally integrated with other attention network regions and with somatomotor regions but decreased in connectivity with DMN (**Supplementary Figure 13**)... The DMN seed showed developmental decreases in connectivity to all other networks while increasing connectivity to other DMN regions (**Supplementary Figure 15**).

And revised Methods section (“Quantification of functional connectivity metrics” subsection) as follows:

Seed-based analyses examining age-related connectivity in exemplar regions from the visual, somatomotor, salience / ventral attention... and default mode networks were conducted using edge-level connectivity data.

5. The results derived from the NKI dataset seem to diverge to some extent from those obtained from other datasets, as shown in Figure 2a and the curve pattern in Figure 5. A discussion exploring the potential factors contributing to these partial discrepancies is recommended.

We agree and are happy to clarify. The sample size used for NKI was significantly smaller than for the other three datasets. Specifically, the sample sizes for all datasets are as follows: PNC, N = 1207; HCP-D, N = 625; HBN, N = 1126, NKI, N = 397. We suspect that this smaller sample size led to greater noise in the estimation of our age effects. Furthermore, NKI was the only dataset that did not acquire fieldmaps; as a result NKI may have been more susceptible to distortions than other datasets, leading to some divergence in NKI results. We now note these two points in our revised Results:

A quantitative analysis confirmed remarkably high consistency in age effects across the four independent datasets. Specifically, spatial Pearson’s correlations between FC strength age effects for each pair of datasets range from 0.49-0.88 (mean correlation = 0.71, $p_{\text{spin}} < 0.01$ for NKI-HBN; $p_{\text{spin}} < 0.0001$ for all other spatial correlations; **Figure 1a**). We note that correlations for NKI may potentially be lower than for other datasets due to a smaller sample size and lack of susceptibility distortion correction in this dataset.

...We found that the across-cortex spatial correlation between fitted FC strength and S-A axis ranks strengthened from age 5 to 22 across all datasets (**Figure 4a-d**), indicating that the spatial patterning of FC strength increasingly resembled the S-A axis with age. Of note, the pattern for the NKI dataset diverges somewhat from that of other datasets, with the alignment to the S-A axis being greater in childhood. However, the pattern of FC strength alignment to the S-A axis converges to an absolute correlation of 0.6,

consistent with other datasets. Differences may be due to NKI having a smaller sample size compared to PNC, HCP-D, and HBN and absence of susceptibility distortion correction, leading to greater noise in the estimation of age effects in younger children.

Minor Issues:

1. In Figure 3, displaying the patterns from all seven of Yeo's networks would be informative. Particularly, insights into the fronto-parietal network, which played a crucial role in maturity prediction in their prior work (Cui et al., *Neuron*, 2020), would be of interest.

We thank the reviewer for this useful suggestion and have included the remaining networks in **Supplementary Figure 3**, copied below.

Supplementary Figure 3. Patterns of maturation across the sensorimotor-association axis for additional functional networks. (a-p) The plots display the developmental trajectories for regions from additional functional networks: (a-d) the visual, (e-h) dorsal attention, (i-l) limbic, and (m-p) fronto-parietal networks. Each line represents an individual region's functional connectivity strength developmental trajectory (zero-centered), modeled using generalized additive models. Colors indicate the rank of a given region along the S-A axis.

We have updated the Results (“Development of functional connectivity varies along the S-A axis” subsection) accordingly:

Developmental trajectories for visual, dorsal attention, limbic, and fronto-parietal networks are displayed in **Supplementary Figure 3**.

2. Line 202: It should be clarified as “the predicted FC strength from GAMs” if the FC strength is indeed predicted from the GAMs.

We appreciate this comment. We have revised the text to clarify that the FC strength is predicted from GAMs. This text is now part of the caption for **Supplementary Figure 7** (originally Figure 1).

In the Philadelphia Neurodevelopmental Cohort (PNC), (a) **the fitted values of functional connectivity strength (FC strength) predicted from regional GAMs** are highest in somatomotor cortices in childhood and increase in these cortical areas with age. In contrast, **fitted** FC strength in association cortices is lower in childhood and tends to decrease with age, resulting in negative FC strength values in transmodal association cortices by early adulthood.

3. Line 288: Are the mentioned five data points outliers? Displaying plots with all data points would be helpful.

We are happy to do this. One data point in PNC, three data points in HCP-D, and one data point in HBN were excluded from the plots in the main text for visualization purposes. We have now included all data points in this figure (now updated from Figure 4 to the new **Figure 3**), copied below.

Figure 3. Development of functional connectivity aligns with the sensorimotor-association axis. (a) The rank of each region in the S-A axis explains the majority of variance in age effects in the Philadelphia Neurodevelopmental Cohort (PNC; $r = -0.71$, $p_{\text{spin}} < 0.0001$). These findings are replicated in additional independent datasets, including (b) Nathan Kline Institute-Rockland Sample (NKI; $r = -0.56$, $p_{\text{spin}} < 0.0001$), (c) Human Connectome Project: Development (HCP-D; $r = -0.62$, $p_{\text{spin}} < 0.0001$), and (d) Healthy Brain Network (HBN; $r = -0.72$, $p_{\text{spin}} < 0.0001$). The age effect of FC strength for each region (Schaefer 200) is plotted against the given region's rank in the S-A axis. Regions that do not display significant change in FC strength over development are colored in gray ($Q_{\text{FDR}} > 0.05$) and were included in the correlation.

4. Line 481: Please clarify what “third” refers to.

We thank the reviewer for pointing out this ambiguity in the text. We have edited this line to improve clarity:

Across datasets, we found that cortices at the sensorimotor **pole** of the S-A axis exhibited marked increases in connectivity through development.

5. Line 755: It appears that some words might be missing.

Thank you. We have revised this line to read as follows:

To calculate the FC strength for a given cortical region, we first computed Pearson correlations of its time series with that of all other regions. We then averaged the Pearson correlations to define FC strength for that region. Hence, FC strength represents the mean edge strength of a given region with all other regions, without thresholding.

6. Line 809: Please elaborate the methodology to generate the S-A ranks

We thank the reviewer for this suggestion and have revised our Methods (“Alignment of functional connectivity metric age effects to the S-A axis” subsection) to further detail how a region’s S-A axis rank was defined:

The S-A axis was previously derived in Sydnor et al. (2021)¹⁹ from multi-modal brain maps that exhibit stereotyped feature variability between primary sensorimotor and transmodal association cortices. As a result, the S-A axis represents the average cortical hierarchy derived from multivariate brain properties. A vertex-level S-A axis (in the fsLR surface-density 32k) was parcellated with study atlases to yield regional S-A axis ranks. The S-A axis is publicly available and was obtained from https://github.com/PennLINC/S-A_ArchetypalAxis for this study.

Specifically, ten cortical maps were used to derive the S-A axis. These ten maps included anatomical hierarchy as quantified by the T1-weighted to T2-weighted ratio,¹⁰⁴ functional hierarchy,⁵ evolutionary hierarchy,¹⁰⁵ allometric scaling,¹⁰⁶ aerobic glycolysis,¹⁰⁷ cerebral blood flow,¹⁰⁸ gene expression,¹⁰⁹ first principal component of NeuroSynth terms,¹¹⁰ externopyramidization,¹¹¹ and cortical thickness.¹⁹ Vertex-level representations for each of the ten maps were obtained. Vertices were rank-ordered based on feature value. Vertex rankings were then averaged across all ten maps to derive the spatial patterning of feature variability along a unidimensional S-A axis. The vertex-level S-A axis was parcellated into five cortical atlases used in this paper (Schaefer 200, Schaefer 400, HCP-MMP, and Gordon) such that each region was assigned a unique S-A rank indicating its relative position on the S-A axis.

References

1. Sydnor, V. J. *et al.* Neurodevelopment of the association cortices: Patterns, mechanisms, and implications for psychopathology. *Neuron* **109**, 2820–2846 (2021).
2. Sydnor, V. J. *et al.* Intrinsic activity development unfolds along a sensorimotor–association cortical axis in youth. *Nat Neurosci* 1–12 (2023) doi:10.1038/s41593-023-01282-y.
3. Luo, A. *et al.* Refinement of Functional Connectivity in Development Aligns with the Sensorimotor to Association Axis. (2022).
4. Buckner, R. L. *et al.* Cortical Hubs Revealed by Intrinsic Functional Connectivity: Mapping, Assessment of Stability, and Relation to Alzheimer’s Disease. *J. Neurosci.* **29**, 1860–1873 (2009).
5. Power, J. D. *et al.* Functional network organization of the human brain. *Neuron* **72**, 665–678 (2011).
6. Rawls, E., Kummerfeld, E., Mueller, B. A., Ma, S. & Zilverstand, A. The resting-state causal human connectome is characterized by hub connectivity of executive and attentional networks. *NeuroImage* **255**, 119211 (2022).
7. Zanto, T. P. & Gazzaley, A. Fronto-parietal network: flexible hub of cognitive control. *Trends Cogn Sci* **17**, 10.1016/j.tics.2013.10.001 (2013).
8. Keller, A. S. *et al.* Hierarchical functional system development supports executive function. *Trends in Cognitive Sciences* **27**, 160–174 (2023).
9. Sepulcre, J. *et al.* The Organization of Local and Distant Functional Connectivity in the Human Brain. *PLOS Computational Biology* **6**, e1000808 (2010).
10. Sarwar, T., Tian, Y., Yeo, B. T. T., Ramamohanarao, K. & Zalesky, A. Structure–function coupling in the human connectome: A machine learning approach. *NeuroImage* **226**, 117609 (2021).
11. Vázquez-Rodríguez, B. *et al.* Gradients of structure–function tethering across neocortex. *Proc Natl Acad Sci USA* **116**, 21219–21227 (2019).
12. Baum, G. L. *et al.* Development of structure–function coupling in human brain networks during youth. *Proc Natl Acad Sci USA* **117**, 771–778 (2020).
13. Siegel, J. S. *et al.* Statistical improvements in functional magnetic resonance imaging analyses produced by censoring high-motion data points. *Human Brain Mapping* **35**, 1981–1996 (2014).
14. Power, J. D., Barnes, K. A., Snyder, A. Z., Schlaggar, B. L. & Petersen, S. E. Steps toward optimizing motion artifact removal in functional connectivity MRI; a reply to Carp. *Neuroimage* **76**, 10.1016/j.neuroimage.2012.03.017 (2013).
15. Ciric, R. *et al.* Benchmarking of participant-level confound regression strategies for the control of motion artifact in studies of functional connectivity. *NeuroImage* **154**, 174–187 (2017).
16. Parkes, L., Fulcher, B., Yücel, M. & Fornito, A. An evaluation of the efficacy, reliability, and sensitivity of motion correction strategies for resting-state functional MRI. *Neuroimage* **171**, 415–436 (2018).
17. Sherman, L. E. *et al.* Development of the Default Mode and Central Executive Networks across early adolescence: A longitudinal study. *Developmental Cognitive Neuroscience* **10**, 148–159 (2014).

REVIEWERS' COMMENTS

Reviewer #1 (Remarks to the Author):

I am satisfied with the authors' responses to my comments.

Reviewer #2 (Remarks to the Author):

The authors have done an excellent job in addressing my queries. I have only two minor suggestions remaining.

Regarding Comment 2

I appreciate the consistency of the age effects. Please also plot in Fig S6 the FCS maps of each processing variation.

Regarding Comment 3

Atlases may vary in region and network sizes, but a key finding from Power et al. (2013, Neuron) was that the FCS of a region depends on the size of the community to which it belongs (i.e., within a given atlas or network partition). Network/community sizes are likely to change through development, and this may influence the age effects. I appreciate that this is a difficult possibility test, but perhaps a statement acknowledging this possibility is sufficient.

Reviewer #3 (Remarks to the Author):

The authors are very responsive and have addressed all my questions. I congratulate the authors for this well executed study.

Manuscript NCOMMS-23-40303-A:

**Functional Connectivity Development along the
Sensorimotor-Association Axis Enhances the Cortical Hierarchy**

We thank the Editors of Nature Communications and the three reviewers of our manuscript for their time and for their valuable feedback. We have addressed the remaining comments from Review 2 below. Modifications to the manuscript based on these comments are indicated in this response letter in blue text.

Reviewer #2 (Remarks to the Author):

The authors have done an excellent job in addressing my queries. I have only two minor suggestions remaining.

Regarding Comment 2

I appreciate the consistency of the age effects. Please also plot in Fig S6 the FCS maps of each processing variation.

We thank the reviewer for this suggestion. We have included the FCS age effect maps of each processing variation below.

Supplementary Figure 6. Sensitivity analyses using absolute correlation, thresholded connectivity matrices, and no global signal regression to compute functional connectivity strength provide convergent results. (a) Original analysis from the PNC correlating the age effect of functional connectivity (FC) strength with sensorimotor-association axis rank is displayed for comparison to results of additional analyses. Sensitivity analyses in the PNC were conducted using FC strength computed (b) using the absolute value of the correlation coefficient as the measure of functional connectivity, (c) from thresholded connectivity matrices (including positive correlations only), and (d) without global signal regression. Consistent with our main findings using global signal regression and Pearson correlation, the development of FC strength varies continuously along the S-A axis. (b-d) The rank of each region in the S-A axis is significantly associated with FC strength age effects in the PNC using absolute correlation ($r = -0.48$, $p_{spin} = 0.00095$), thresholded matrices ($r = -0.49$, $p_{spin} = 0.0032$), and no global signal regression ($r = -0.59$, $p_{spin} = 0.0001$). In association regions at the top of the S-A axis (primarily in the default network) in (b) and (c), negative age effects are abolished and are found to be positive. The age effect of FC strength remains negatively associated with S-A axis rank even after excluding GSR (d), suggesting that GSR

played a minimal role in our findings. Maps of mean functional connectivity strength across all ages are displayed for (e) the original analysis in PNC and analyses (f) using the absolute value of the correlation coefficient as the measure of functional connectivity, (g) using thresholded connectivity matrices, and (h) excluding global signal regression. Spearman's rank correlations were used to quantify the association between S-A axis ranks and observed developmental effects with statistical significance determined using spin-based spatial permutation tests.

Regarding Comment 3

Atlases may vary in region and network sizes, but a key finding from Power et al. (2013, Neuron) was that the FCS of a region depends on the size of the community to which it belongs (i.e., within a given atlas or network partition). Network/community sizes are likely to change through development, and this may influence the age effects. I appreciate that this is a difficult possibility test, but perhaps a statement acknowledging this possibility is sufficient.

We thank the reviewer for this comment. We have added the following limitation to our Discussion:

Furthermore, because we utilized standardized group average cortical parcellations and network solutions, we were not able to evaluate whether network or community sizes changed through development.